



# Objective extraction and analysis of statistical features of Dansgaard-Oeschger events

Johannes Lohmann[1] and Peter D. Ditlevsen[1]

[1]Centre for Ice and Climate, Niels Bohr Institute, University of Copenhagen, Denmark

**Correspondence:** Johannes Lohmann (johannes.lohmann@nbi.ku.dk)

**Abstract.** The strongest mode of centennial to millennial climate variability in the paleoclimatic record are the Dansgaard-Oeschger (DO) cycles. Despite decades of research their dynamics and physical mechanism remain poorly understood. Valuable insights can be obtained by studying high-resolution Greenland ice core proxies, such as the NGRIP $\delta^{18}$O record. However, conventional statistical analysis is complicated by the high noise level, the cause of which is partly due to glaciological effects

unrelated to climate, and which is furthermore changing over time. We remove the high-frequency noise and extract the most robust features of the DO cycles, such as rapid warming and interstadial cooling rates, by fitting a consistent piecewise-linear model to three Greenland ice core records. With statistical hypothesis tests we aim to obtain an empirical, mechanistic understanding of what controls the of amplitudes and durations of the DO cycles. To this end, we investigate distributions of and causalities in between different features, as well as modulations of them in time by external climate factors, such as $CO_2$ and

insolation. Our analysis suggests different mechanisms underlying warming and cooling transitions due to contrasting distributions and external influences of the stadial and interstadial durations, as well as the fact that the interstadial durations can be predicted to some degree by the linear cooling rates already shortly after interstadial onset.

## 1    Introduction

The physical mechanism(s) and cause of the Dansgaard-Oeschger (DO) events are largely unknown and debated. Modeling

and simulations of the events are guided by the proxy records, among which the stable water isotope records from Greenland ice cores are very prominent. The records are noisy, and since there are no established theories about how they should evolve, there is no obvious filter to extract the large-scale climate signal. A common characteristic of the DO cycles seems to be an abrupt temperature increase from the cold stadial conditions to a maximum temperature in the warm interstadial state followed by a gradual cooling until there is another abrupt jump back into the stadial state. This is referred to as the saw-tooth shape of

the events.

Due to the high noise level in the record it is however difficult to discern this specific structure in all of the events. Some events do not seem to follow the generic shape. Furthermore, there are very short events where it is difficult to speak of a gradual cooling episode. Even other events are interrupted by shorter cooling episodes, referred to as sub-events (Capron et al., 2010). As interpretation of noisy time series are often biased, subjective and one is prone to recognize patterns that can arise by

chance, we seek a quantitative evaluation of the record. Assuming the saw-tooth shape of the events, we develop an algorithm



for fitting the saw-tooth shape to the entire NGRIP $\delta^{18}$O record of the last glacial, similar to ramp-fitting a jump in a noisy record.

Firstly, our method gives an objective basis of the validity of the generic saw-tooth description of the DO cycles and identify which individual cycles fall outside this description. Secondly, with a piecewise-linear fit, we obtain estimates for the stadial and interstadial levels, the abruptness of the transitions and the gradual cooling rate in the interstadial periods. By bootstrapping, we estimate the uncertainty in extracting these parameters from the noisy background. Finally, we perform a statistical analysis of the fit parameters across the DO events of the last glacial period, in order to obtain a mechanistic understanding of what controls the evolution of amplitudes and durations of the DO cycles. This can potentially be used for identifying or excluding proposed mechanisms and for bench-marking model results.

Previous efforts to extract robust DO event features from the record were conducted on only part of the record and were focused on single or very few features. In Schulz (2002), linear fits to the interstadials were used to infer the cooling rates starting with Greenland interstadial 14 (GI-14). Estimates for abruptness of warming transitions and durations of interstadials have been derived by Rousseau et al. (2017), starting at GI-17.1. A comprehensive survey of onset times of all interstadial and stadial periods is given in Rasmussen et al. (2014). Our work is different in that we derive all features that can be extracted from a saw-tooth shaped fit to all events at once, by using a fit that is consistent and continuous throughout the record. We thus do not have to rely on any subjective choice of stadial and interstadial onsets or levels. We do, however, not attempt to define the DO events themselves from the record, but instead use the fixed set of all previously classified events (Rasmussen et al., 2014).

In this study, we show that a characteristic saw-tooth waveform can be fit to all DO cycles. However, almost half of the cycles do not actually display a significant rapid cooling episode after the more gradual interstadial cooling. A subsequent statistical analysis of the DO event features hints at different mechanisms underlying warming and cooling transitions. Specifically, we find different distributions of the stadials and interstadial durations, as well as different external factors that could influence them. Furthermore, the interstadial durations can be predicted to some degree by the linear cooling rates within a few hundred years after interstadial onset. In contrast, the stadial and rapid warming durations are consistent with the stadial-interstadial transitions as spontaneous, noise-induced escapes from a metastable state.

The paper is structured in the following way: In Sec. 2 we introduce the data used in the study and its pre-processing, the iterative fitting algorithm, the features we extract from the saw-tooth shape fit to the events and the statistical tools to analyze these features. In Sec. 3 we report the results of the fit. Section 3.1 discusses the appropriateness of the saw-tooth fit to the events, and Sections 3.2 and 3.3 treat the uncertainty in estimating the fit parameters and the derived features. In Sec. 4 we analyze in detail the features characterizing the stadial, interstadial and abrupt warming periods. The results of the fit and the implications of the subsequent data analysis are discussed in Sec. 5. We conclude in Sec. 6.



## 2 Methods and Materials

### 2.1 Data

The basis of our study is the $\delta^{18}O$ Greenland ice core record of the last glacial period (120 - 12 kyr BP, kyr BP = one thousand years before present). In the NGRIP ice core, $\delta^{18}O$ has been measured in 5 cm samples (NGRIP Members, 2004; Gkinis et al.,

2014; Rasmussen et al., 2014). These raw depth measurements are transferred to the GICC05 time scale (Svensson et al., 2006). This results in an unevenly spaced time series with a resolution of 3 years at the end to 10 or more years at the beginning of the last glacial period. Because it greatly simplifies our analysis, we transfer this to an evenly-spaced time series by oversampling to 1 year resolution using nearest-neighbor interpolation. Thus, we do not alter the actually measured values, adding or removing any variability. For subsequent comparison, the high-resolution $\delta^{18}O$ record from the GRIP ice core on the GICC05

time scale has been used (Johnsen et al., 1997; Rasmussen et al., 2014), and processed in the same way. Our fit is performed on a previously classified set of events from Greenland, which has been reported by Rasmussen et al. (2014) together with the time stamps. These time stamps are used to initialize our iterative fitting procedure, and are subsequently refined during the process. We do not treat sub-events, which are small dips to colder conditions during a warm period, as separate events, but instead fit them as part of the interstadial periods.

In our data analysis, we use several other data sets that are not derived from Greenland ice cores. These are loosely referred to as external forcings, although not all are truly external to the climate system, but rather obtained from independent data sources. As proxy for global ice volume, we use the LR04 ocean sediment stack (Raymo and Lisiecki, 2005). To represent Antarctic temperatures, we choose the $\delta^{18}O$ record of the EDML ice core on the AICC12 time scale (EPICA Community Members,

2010). This data was processed by interpolation to an equidistant 20 year grid and subsequently smoothing by convolution with a 600 year Hamming window. Greenhouse gas forcing is represented by a composite $CO_2$ record from different Antarctic ice cores on the AICC12 gas time scale (Bereiter et al., 2015). Furthermore, we consider changes in insolation due to orbital variations. Firstly, we use incoming solar radiation at 65 degree North integrated over the summer (referred to as 65Nint hereafter), which we define as the annual sum of the radiation on days exceeding an average of 350 W/m$^2$ (Huybers, 2006).

Secondly, we use incoming solar radiation at 65 degree North at summer solstice (referred to as 65Nss hereafter) (Laskar et al., 2004). In addition, we also consider the raw orbital parameters of obliquity, eccentricity, and precession index (Laskar et al., 2004).

### 2.2 Fitting routine

A naive approach to obtain a piecewise-linear fit of the whole record could proceed in the following way: Considering the

stadials as constant, first cut the time series at a predefined beginning and end of two consecutive stadials. Then fit a saw-tooth shape to the event within the two stadials. The end of the fit to this event then determines the start of the next stadial, used to fit the following event. However, the point at which one initially cut the stadials influences the constant levels of the two stadials that have been used to determine the fit to the event. For a consistent fit, the start and end points of a stadial must rather





be determined by the fits to two neighboring events. In this way, the fit to each event depends on both its neighboring events before and after, and we cannot fit the events sequentially. One solution is to fit the whole time series at once to a piecewise linear model with 186 parameters, corresponding to 6 parameters for each of the 31 DO events. However, due to high noise and abundance of sub-event features, such a fit will be difficult to achieve without invoking very complicated constraints.

Instead, we propose an iterative fitting routine that converges to a consistent fit, as detailed in the following. We start with a guess for the stadial onset and end times, which determine the constant stadial levels. Then we fit a saw-tooth shape individually to each event. Thereafter, we update the stadial onset and end times according to the fit and repeat the procedure. When after some iterations the onset and end times do not change significantly anymore the fit has converged and is consistent.

We start with an initial guess of the onsets and ends of the stadial periods, based on the timings reported by Rasmussen et al. (2014), which are kept fixed throughout the iterations of our algorithm. The time series is divided in segments at these times. For each event $i$, we take a segment consisting of a stadial and interstadial period plus the following stadial period. These segments are fitted individually to a piecewise-linear model, as shown in Fig. 1. The model starts with a constant line at the beginning of the stadial. The constant is fixed to the mean level of the stadial $l_i^s$, where the stadial beginnings and ends

are determined by the initial guess, or the previous iteration. A first break point (parameter $b_1$) of this constant is determined, followed by a linear up-slope (parameter $s_1$). The slope ends at the second break point (parameter $b_2$). After this break point there is a linear down-slope (parameter $s_2$), which ends at a break point (parameter $b_3$). After this break point there is a steeper down-slope until a last break point (parameter $b_4$), which is at the level of the next stadial $l_{i+1}^s$ that is determined from the previous iteration. After all events have been fit, the parameters $b_4$ and $b_1$ of each event update the beginnings and ends of

the stadials. The updated stadials yield a new segmentation of the time series and new stadial levels, which are then used as constant segments in the next iteration. The idea of this approach is that if the problem is well behaved, the beginnings and ends of the stadials do not change significantly anymore after a certain number of iterations, meaning that a consistent fit of the entire time series is obtained. An algorithm for this routine, along with details of the optimization procedure to fit each event, is given in Appendix A.

The fitting procedure outlined above yields a single best fit that we hope to be close to the absolute global minimum of the optimization problem and furthermore as consistent as possible, meaning that the stadial sections that were used for the fit in the last iteration are identical to the stadial sections defined by the resulting fit. Additionally to this best fit we assess the uncertainty in each of the parameters that arise due to noise in the record. We cannot estimate this from the output of our fitting procedure in a straightforward way. Instead, we use bootstrapping to repeatedly generate synthetic data for each transition

and optimize the parameters with basin-hopping. Like this, we yield a distribution on each parameter. Due to computational demands, we do not combine this with our iterative procedure, but rather resample and fit every transition independently. Thus, we neglect the co-variance structure of the errors in the parameters of neighboring transitions. However, we still consider it to be a very good estimate of the uncertainty due to the noise in the record. The detailed procedure is given in Appendix C.



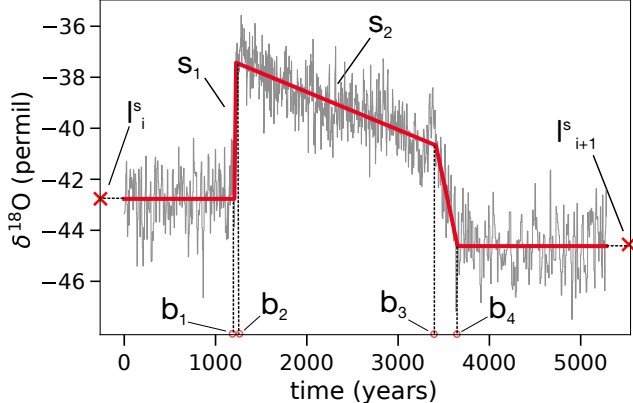

**Figure 1.** Piecewise-linear model fit to DO event 20, where the time series consists of GS-21.1, GI-20 and GS-20. The parameters of the piecewise-linear model are the four break-points $b_{1,2,3,4}$, the up-slope $s_1$ and the down-slope $s_2$. The constant levels $l_i^s$ and $l_{i+1}^s$ of GS-21.1 and GS-20 are constant during an iteration of the fitting routine, and are updated when after each iteration all breakpoints have been determined.

## 2.3 DO event features

From the best fit parameters of each DO cycle a variety of features follow, which form the core of our analysis. For each rapid warming period, gradual interstadial cooling period, as well as rapid cooling period at the end of an interstadial, we consider the duration, rate of change and the amplitude. Furthermore, several absolute levels are of interest, including the constant stadial
5    levels, the interstadial levels after the abrupt warming and the interstadial level before the rapid cooling. As a level relative to each event, we consider the level before the rapid cooling above the previous stadial level, which is given by the rapid warming amplitude minus the gradual cooling amplitude. Finally, the gradual cooling amplitude divided by the rapid warming amplitude measures the position of the point of rapid cooling within the event amplitude. In total, we consider 15 interdependent climate features, which are listed in Tab. 1.

## 2.4 Data analysis

Our aim is to develop a mechanistic understanding of the evolution of the DO cycles. To this end, we employ several tools to search for relations between different features, as well as between features and external climate factors. Because of the large number of possible combinations of features, we first pre-select significant and potentially relevant relationships and thereafter
15    investigate in detail, whether the results are robust to outliers, among other things. In some cases we also consider relationships of features and forcing that are not significant for the whole data set, but for a large subset. This might highlight that there were qualitatively different periods within the last glacial, or that some DO events are of different nature than most.



**Table 1.** List of DO event features obtained from the fit that are analyzed in this study.

| Feature | Definition |
|---|---|
| Warming duration | $b_2 - b_1$ |
| Warming rate | $s_1$ |
| Warming amplitude | $s_1(b_2 - b_1)$ |
| Gradual cooling dur. | $b_3 - b_2$ |
| Gradual cooling rate | $-s_2$ |
| Gradual cooling ampl. | $s_2(b_2 - b_3)$ |
| Fast cooling dur. | $b_4 - b_3$ |
| Fast cooling rate | $\frac{s_1(b_2-b_1)+s_2(b_3-b_2)-(l^s_{i+1}-l^s_i)}{(b_4-b_3)}$ |
| Fast cooling ampl. | $s_1(b_2 - b_1) + s_2(b_3 - b_2) - (l^s_{i+1} - l^s_i)$ |
| Stadial duration | $b_1$ |
| Stadial level | $l^s$ |
| Interstadial level | $s_2(b_2 - b_3) + l^s$ |
| Interstadial end level | $s_1(b_2 - b_1) + s_2(b_3 - b_2) + l^s$ |
| Relative Int. end level | $s_1(b_2 - b_1) + s_2(b_3 - b_2)$ |
| Cooling/warming ampl. | $s_2(b_2 - b_3) \cdot [s_1(b_2 - b_1)]^{-1}$ |

We first consider Pearson and Spearman correlation coefficients of pairs of features and external climate factors. We pre-select combinations with p-values $p < 0.1$, which are estimated by permutation tests that assume independent samples. For a given number of data points in a sequence, the true p-values should often be higher due to autocorrelation. Along with other potential artifacts, this is investigated individually for the pre-selected combinations.

Next, in order to find relations between more than two variables, we search for multiple linear regression models to explain selected features of the data. For this, we often use logarithmic quantities because with many features it is otherwise unlikely to find linear relationships that are not dominated by outliers. Given a feature as response variable, we fit linear regression models of combinations of two other features or forcings and pre-select models with the largest coefficients of determination, in order to further analyze the fit.

Furthermore, in order to find subsets of events that have distinct properties or causal relations that are only valid in part of the data, we perform a clustering analysis on the data, using two different algorithms (K-means and Agglomerative Hierarchical Clustering). Given our sample size of 31 events, we search for clusterings with 2 or 3 clusters. Potentially relevant clusterings are pre-selected by considering the mean Silhouette coefficient, which is a distance-based measure for the validity of clusters. With the abovementioned tools, we perform an analysis on the entire set of features and forcings. From the results obtained,

we report selected findings, which are most robust and relevant, in Sec. 4 of this paper.

It is important to be aware of the problem of multiple comparisons when interpreting the significance of such an analysis. Tests for significant correlations of many pairs of features using, e.g., the Spearman correlation coefficient, yield a non-negligible number of false positives when using confidence levels that are reasonable for our purposes. We consider both features of the same and neighboring events, yielding $\frac{15 \cdot 14}{2} = 105$ and $15 \cdot 15 = 225$ tests, respectively. Furthermore, we test

the correlation of all features and forcings, yielding another $15 \cdot 8 = 120$ tests. Assuming these are all independent tests, the



total expected number of false positives is 22.5 at 95% confidence, while at 99% it is 4.5. Since we derive 15 features from only 7 independent parameters (including stadial levels) for each DO cycle, many pairs of features within the same DO cycle are related and thus we expect true positives for correlation tests. For instance, this is true for warming amplitude and interstadial level, or relative interstadial level and gradual cooling amplitude. Similarly, due to the constraints on the fit parameters, the

rates and durations of fast and gradual cooling are correlated. These types of correlations are not relevant and thus this reduces the number of pairwise correlations to consider. For combinations of amplitude, duration and rates of a given linear segment we also expect correlation, because they are trivially related: duration = amplitude $\cdot$ (rate)$^{-1}$. However, it is interesting to test whether the rates or the amplitudes more strongly determine the durations, which essentially depends on which of the two has a larger variability. We investigate this for the different periods of the DO cycles below.

There are sophisticated methods to control the multiple comparisons problem. These could be helpful to better detect false positives from our analysis, but depend on being able to properly estimate the significance of individual correlations in between features with autocorrelation and assess the statistical dependence of the hypothesis tests due to the dependence of some of the features. For simplicity, we do not consider such an analysis, but consider individually significant correlations as suggestions to be investigated further.

**3    Piecewise-linear fit of NGRIP record**

The iterative fitting routine is performed for 40 iterations, so that the initial fluctuations in the parameters have died out and converged to a consistent fit, as detailed in Appendix B. In Fig. 2, the resulting fit is superimposed onto the high-resolution NGRIP time series. We fit 31 DO events in total, starting with DO 24.2 and ending at DO 2.2, excluding the two outermost events of the last glacial, because they are very non-stationary in their stadial parts. Table 2 shows all parameters obtained from

the fit. Instead of $b_{1,2,3,4}$ for each transition, we show the corresponding times of stadial end, interstadial onset, interstadial end and stadial onset.

**3.1    Saw-tooth shape of DO events**

In our fit, all transitions follow the characteristic saw-tooth shape. For a few events, this is because of the constraints we use in the fitting algorithm. Typically, the constraints do not strictly bound the best fit parameters, but they force the fit into

another local minimum that is consistent with the saw-tooth shape, which often yields parameters that are still clearly within the constraints. There are, however, four events where the best fit parameters actually lie very close to the bounds set by the constraints. This happens for GI-5.1 and GI-3, which both have ratios of rapid to gradual cooling rates very close to the constraint value of 2.0. Similarly, for GI-15.2 and GI-6 the ratio of gradual to rapid cooling duration is close to 2.0. Detailed pictures of each transition and the corresponding fit are shown in Fig. S2 in the supplementary material.

The fact that some constraints are needed in order to ensure that the fit of each event follows a saw-tooth shape can be used to classify which events fall outside of this description. To this end, we perform another run of the iterative fitting routine without using constraints 3, 4, 6 and 7 listed in Appendix A. From the resulting fit we then analyze, which of the events are





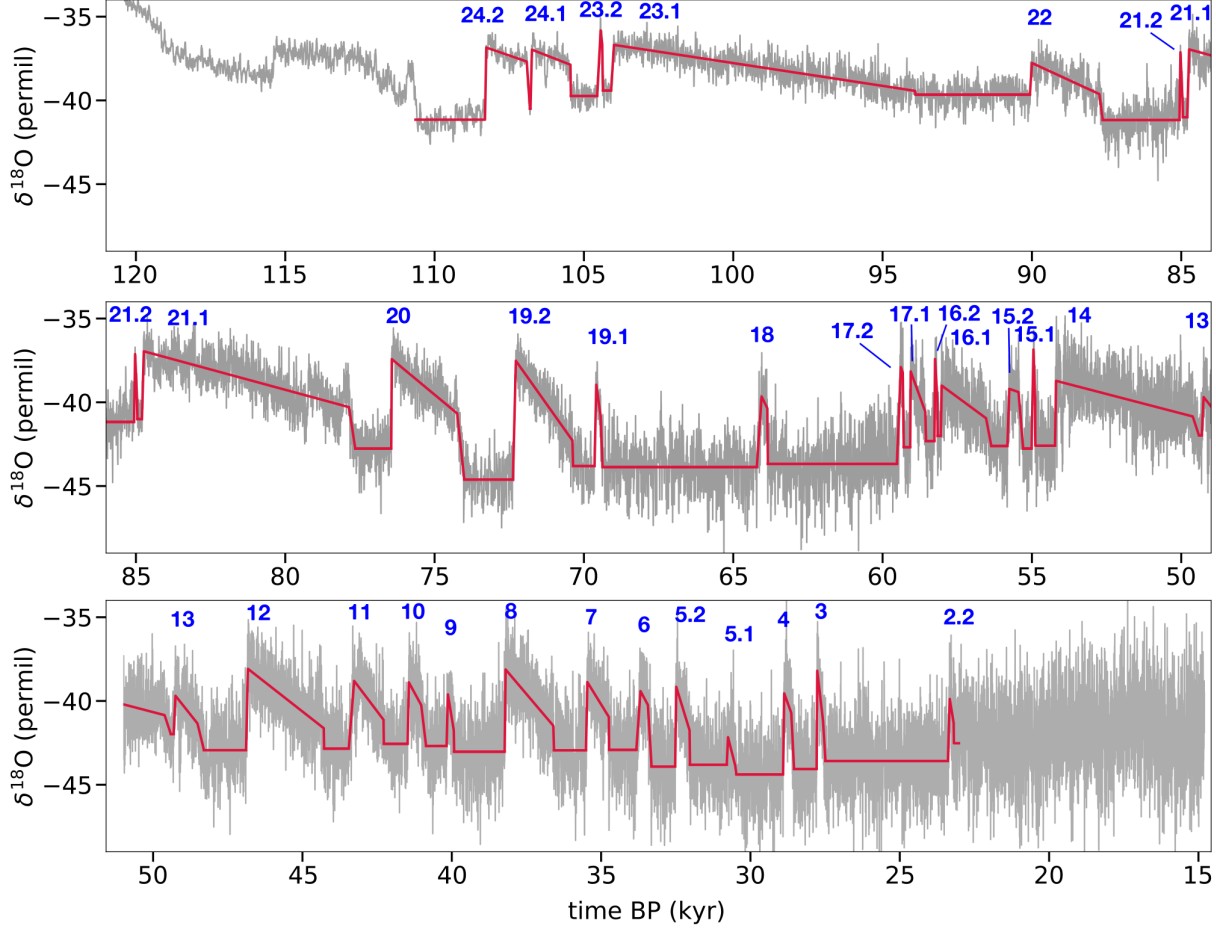

**Figure 2.** High-resolution NGRIP $\delta^{18}$O time series and the piecewise-linear fit obtained by our method. The numbers above the interstadials indicate the names of the DO cycles considered in this study.

not consistent with the saw-tooth shape. For this, we use 4 criteria: 1. The abrupt cooling rate is at least twice as large as the gradual cooling rate. 2. The gradual cooling lasts at least twice as long as the abrupt cooling. 3. There is gradual cooling after the rapid warming, i.e., the gradual cooling rate is negative. 4. The abrupt cooling amplitude is larger than 0.5 permil. Criterion 1 is not met by events 23.1, 19.2, 15.1, 11, 5.1, 3 and 2.2, criterion 2 by events 21.2, 19.2, 17.2, 15.2, 15.1, 11, 10, 9, 8, 6, 5.1,

5   3 and 2.2, criterion 3 by event 11, and criterion 4 by events 23.1 and 15.1. By demanding that all of these criteria are met, we thus conclude that the following 14 out of 31 events fall outside of the saw-tooth description: 23.1, 21.2, 19.2, 17.2, 15.2, 15.1, 11, 10, 9, 8, 6, 5.1, 3 and 2.2.



**Table 2.** Parameters resulting from the fitting routine on the NGRIP data.

| Event | Stadial End (yr BP) | Interstadial Onset (yr BP) | Interstadial End (yr BP) | Stadial Onset (yr BP) | Warming Rate (permil/yr) | Cooling Rate (permil/yr) |
|---|---|---|---|---|---|---|
| 24.2 | 108313 | 108270 | 106914 | 106810 | 0.0992 | 0.00062 |
| 24.1 | 106790 | 106743 | 105452 | 105439 | 0.0744 | 0.00069 |
| 23.2 | 104556 | 104441 | 104387 | 104366 | 0.0340 | 0.01555 |
| 23.1 | 104090 | 103996 | 93916 | 93898 | 0.0290 | 0.00027 |
| 22 | 90069 | 89999 | 87743 | 87631 | 0.0270 | 0.00082 |
| 21.2 | 85060 | 85027 | 84964 | 84952 | 0.1230 | 0.03992 |
| 21.1 | 84799 | 84737 | 77866 | 77659 | 0.0655 | 0.00049 |
| 20 | 76452 | 76434 | 74245 | 74009 | 0.2935 | 0.00148 |
| 19.2 | 72377 | 72280 | 70385 | 70365 | 0.0730 | 0.00251 |
| 19.1 | 69646 | 69587 | 69443 | 69381 | 0.0831 | 0.01262 |
| 18 | 64212 | 64051 | 63858 | 63846 | 0.0262 | 0.00367 |
| 17.2 | 59520 | 59390 | 59323 | 59294 | 0.0446 | 0.00503 |
| 17.1 | 59076 | 59061 | 58571 | 58549 | 0.2951 | 0.00491 |
| 16.2 | 58266 | 58245 | 58168 | 58162 | 0.2340 | 0.03107 |
| 16.1 | 58051 | 58023 | 56536 | 56364 | 0.1059 | 0.00131 |
| 15.2 | 55821 | 55759 | 55449 | 55296 | 0.0554 | 0.00062 |
| 15.1 | 55011 | 54950 | 54892 | 54887 | 0.0981 | 0.05104 |
| 14 | 54228 | 54193 | 49617 | 49410 | 0.1092 | 0.00046 |
| 13 | 49315 | 49253 | 48517 | 48301 | 0.0367 | 0.00223 |
| 12 | 46890 | 46826 | 44286 | 44277 | 0.0761 | 0.00140 |
| 11 | 43450 | 43271 | 42285 | 42278 | 0.0225 | 0.00236 |
| 10 | 41479 | 41439 | 41024 | 40864 | 0.0910 | 0.00326 |
| 9 | 40175 | 40131 | 39933 | 39928 | 0.0699 | 0.01096 |
| 8 | 38231 | 38199 | 36602 | 36583 | 0.1549 | 0.00210 |
| 7 | 35508 | 35461 | 34741 | 34735 | 0.0859 | 0.00289 |
| 6 | 33822 | 33681 | 33434 | 33314 | 0.0250 | 0.00334 |
| 5.2 | 32521 | 32485 | 32039 | 32028 | 0.1324 | 0.00583 |
| 5.1 | 30794 | 30752 | 30514 | 30473 | 0.0394 | 0.00695 |
| 4 | 28908 | 28871 | 28635 | 28544 | 0.1302 | 0.00485 |
| 3 | 27786 | 27765 | 27572 | 27492 | 0.2762 | 0.01529 |
| 2.2 | 23389 | 23328 | 23196 | 23191 | 0.0607 | 0.01098 |

## 3.2 Uncertainty of fit parameters and features

From the best fit, we estimate the uncertainty of each parameter via bootstrapping, as explained in Appendix C. As an example, we show distributions of the parameters for DO event 20 in Fig. 3. In Tab. 3 we show the durations and amplitudes of the rapid warmings for each event along with a bootstrap confidence interval consisting of the 16- and 84-percentiles, which would

5   correspond to the $\pm\sigma$ range if the distributions were Gaussian. The actual distributions are often skewed, so that the best fit values lie close to the edges of the confidence intervals, or even outside of the intervals. In these cases, the $\pm\sigma$ confidence intervals are not the best indicator for the uncertainty, because they barely include the mode of the very skewed distributions.




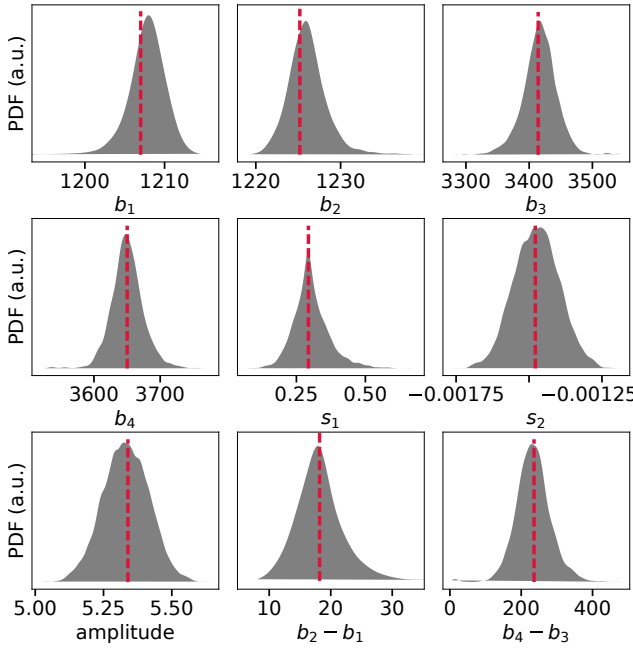

**Figure 3.** Gaussian kernel density of the density of model parameters and some derived quantities for the DO event 20 after 5000 iterations of the bootstrap resampling procedure. The parameter values for the best fit, as reported in Sec. 3, are indicated with *red dashed* lines. The amplitude feature is given by $s_1(b_2 - b_1)$.

The magnitude of the uncertainty varies from event to event. In the case of the warming durations, the average bootstrap standard deviation is 20.0 years, with a minimum of 3.4 years for GI-16.2 and a maximum of 57.4 years for GI-18. We observe that shorter warmings typically also have smaller uncertainties. As comparison, the durations of the rapid coolings at the end of an interstadial have a larger uncertainty with an average bootstrap standard deviation of 53.6 years. This is expected, because the rapid cooling is typically less well pronounced in the record compared to the rapid warming. The coolings also have a larger spread in the bootstrap standard deviations with a minimum of 4.6 years for GI-16.2 and a maximum of 209.9 years for GI-23.1, because some events have a very clearly defined rapid cooling, while others do not. Similarly, the onset times of the rapid warmings have an average bootstrap standard deviation of 11.4 years, whereas the onset of the stadial periods have a corresponding average uncertainty of 31.7 years.

## 3.3   Comparison of NGRIP and GRIP records

As complementary approach to assess the uncertainties of the features, we compare them to those derived in the same way from another Greenland ice core. We chose the $\delta^{18}$O record of the GRIP ice core, which is measured at a similar resolution to the NGRIP record and has been transferred to the GICC05 time scale starting at the onset of GI-23-1. We thus started fitting the record from GS-22 with 40 iterations of our algorithm, using the same constraints and hyperparameters. Again, the algorithm





**Table 3.** Durations and amplitudes of the rapid warmings inferred from the fit, together with a confidence interval obtained by bootstrapping.

| Event | Warming duration (yr) | | | Amplitude (permil) | | |
|---|---|---|---|---|---|---|
| | Best fit | 16-p | 84-p | Best fit | 16-p | 84-p |
| 24.2 | 43.4 | 36.3 | 47.8 | 4.30 | 4.23 | 4.40 |
| 24.1 | 47.4 | 34.9 | 45.0 | 3.53 | 3.42 | 3.61 |
| 23.2 | 115.2 | 96.1 | 126.1 | 3.92 | 3.72 | 4.11 |
| 23.1 | 94.1 | 78.9 | 127.3 | 2.73 | 2.69 | 2.75 |
| 22 | 70.0 | 70.3 | 91.8 | 1.89 | 1.78 | 1.95 |
| 21.2 | 33.0 | 25.7 | 39.9 | 4.06 | 3.61 | 4.10 |
| 21.1 | 61.7 | 53.5 | 79.6 | 4.05 | 3.98 | 4.09 |
| 20 | 18.2 | 14.7 | 21.6 | 5.34 | 5.25 | 5.42 |
| 19.2 | 97.2 | 74.3 | 98.1 | 7.09 | 6.93 | 7.19 |
| 19.1 | 58.5 | 37.7 | 60.0 | 4.86 | 4.45 | 4.97 |
| 18 | 161.0 | 74.6 | 194.0 | 4.21 | 3.99 | 4.51 |
| 17.2 | 129.7 | 83.7 | 158.0 | 5.79 | 5.47 | 6.20 |
| 17.1 | 15.3 | 13.9 | 27.0 | 4.53 | 4.14 | 4.72 |
| 16.2 | 21.0 | 18.6 | 24.0 | 4.92 | 4.59 | 5.19 |
| 16.1 | 28.4 | 28.9 | 84.0 | 3.01 | 2.88 | 3.16 |
| 15.2 | 61.6 | 39.0 | 100.0 | 3.41 | 3.38 | 3.67 |
| 15.1 | 60.6 | 56.4 | 69.2 | 5.94 | 5.68 | 6.12 |
| 14 | 35.5 | 38.0 | 79.0 | 3.87 | 3.78 | 3.95 |
| 13 | 62.4 | 63.4 | 101.2 | 2.29 | 2.07 | 2.60 |
| 12 | 63.9 | 45.7 | 73.8 | 4.86 | 4.71 | 4.94 |
| 11 | 179.5 | 143.0 | 201.0 | 4.05 | 3.86 | 4.17 |
| 10 | 40.3 | 41.3 | 80.2 | 3.67 | 3.40 | 3.97 |
| 9 | 44.2 | 37.5 | 86.4 | 3.09 | 2.66 | 3.22 |
| 8 | 31.7 | 29.8 | 53.0 | 4.91 | 4.78 | 4.98 |
| 7 | 47.4 | 45.3 | 90.2 | 4.07 | 3.86 | 4.27 |
| 6 | 140.4 | 110.6 | 172.1 | 3.51 | 3.41 | 3.92 |
| 5.2 | 36.0 | 31.1 | 54.6 | 4.76 | 4.45 | 4.93 |
| 5.1 | 41.8 | 41.4 | 82.0 | 1.65 | 1.51 | 1.89 |
| 4 | 37.2 | 27.1 | 41.8 | 4.84 | 4.47 | 5.11 |
| 3 | 21.3 | 18.0 | 25.0 | 5.88 | 5.40 | 5.92 |
| 2.2 | 61.2 | 42.0 | 91.6 | 3.72 | 3.21 | 3.75 |

converges to a consistent fit, where each of the events is well approximated by a saw-tooth shape. We now describe how well the features of NGRIP and GRIP correspond for the 26 mutual events.

For the gradual cooling rates, the Pearson (Spearman) correlation is $r_p = 0.64$ ($r_s = 0.65$). Here, the discrepancy in between the two records is only due to a handful of short events that show a clear linear cooling slope in one record, but are more

5 plateau-like in the other. This happens for the interstadials 18, 16.2 and 5.1, which don't show a slope in GRIP, and 17.2, which doesn't show a strong slope in NGRIP. If we remove these events, the correlation is $r_p = 0.97$ and $r_s = 0.98$. The warming durations show a correlation of $r_p = 0.55$ and $r_s = 0.63$. There are no outliers, but a rather large spread, indicating that the warming duration is a less robust feature compared to the cooling rate. With 69 years on average, the GRIP warmings are 8 years shorter than the NGRIP average. The average absolute deviation of warming durations in the two cores is 31 years, with

10 a maximum discrepancy of 103 years for GI-10, where we find a warming of 40 years for NGRIP and 143 years for GRIP.



Such deviations can arise if there is a slight step in the record before the most rapid warming and the algorithm includes this in the fit.

The warming amplitudes are very well correlated with $r_p = 0.87$ and $r_s = 0.83$. The average amplitude of 3.87 permil in GRIP is 0.45 permil lower than the NGRIP average. The stadial levels are also well correlated with $r_p = 0.78$ and $r_s = 0.66$.

There is a quite consistent offset in between the cores of 1.84 permil due to differences in altitude and latitude of the GRIP and NGRIP sites. Exceptions include GS-21.1, which does not follow the offset but is at a very similar level in both GRIP and NGRIP, and GS-14, which is difficult to define and thus vulnerable to give different results due to different noise in the cores.

The rapid cooling durations, i.e. $b_4 - b_3$, show an average absolute deviation in between the two cores of 59 years, with $r_p = 0.46$ and $r_s = 0.53$. This corroborates the fact that this feature is harder to define than the rapid warmings. The break

points $b_3$ and $b_4$ are very susceptible to noise and can yield qualitatively different results for different cores. As a result, the abrupt cooling of GI-19.2 lasts 208 years in GRIP and only 20 years in NGRIP, and for GI-12 294 years in GRIP and only 9 years in NGRIP. Conversely, the abrupt coolings of GI-19.1, GI-10 and GI-6 last much longer in NGRIP, with 62, 160 and 120 years in NGRIP versus 2, 5 and 2 years in GRIP, respectively. Consequently, we do not report any results concerning the rapid cooling period in this paper.

The stadial and interstadial durations are very well correlated with $r_s = 0.99$ and $r_s = 0.97$, respectively. The average absolute deviation is 59 years for interstadials and 73 years for stadials, which is small compared to the average durations. The biggest discrepancies in between the two cores come from the indeterminacy in the rapid coolings of certain events, as described above.

In summary, the uncertainties obtained by bootstrapping and by comparison with the GRIP ice core are compatible. The

average bootstrap standard deviation of rapid warming and cooling durations is 20 and 54 years, respectively. This compares well to the average absolute deviation in between GRIP and NGRIP of warming and cooling durations of 31 and 59 years, respectively. The discrepancy of 31 years for warming durations also includes a systematic bias of on average 8 year longer warmings in GRIP. Thus the unbiased uncertainty is likely even closer to the one obtained by bootstrapping. Shorter time scale features like rapid warming durations are not fully representative for every single event in one core. However, the overall trends

are consistent, as seen by significant correlation. Features on longer time scale, such as most of the cooling slopes and stadial levels, are clearly representative. The same holds for stadial and interstadial durations.

## 4 Statistical analysis of DO event features

In Fig. 4 we show histograms of all the DO event features derived from the fit parameters that we consider in this study, as defined in Sec. 2.3. From the histograms we can see that the features have different types of distributions. Whether an event

feature should be considered an independent sample from a distribution depends on whether it shows a significant trend over time. If we consider the event-wise sequence of one feature as evenly spaced time series we can calculate the autocorrelation and determine by a permutation test, whether the value at a given lag is significantly larger than what could be expected in an uncorrelated sample for a given confidence. By considering autocorrelations up to lag 5, we find that the three different





levels (stadial, interstadial and level before rapid cooling) show significant autocorrelation at 95% confidence until a lag of 3. We also find significant autocorrelation for four other features at only one specific lag value each, which we believe are false positives. In fact, when independently testing the hypothesis of significant autocorrelation at 95% confidence for 15 different time series (features) at 5 lags, there is an expected value of 3.75 false positives. The corresponding data is shown in Fig. S3

5    in the Supplementary material. As a result, in most cases we can consider the features of each event as independent samples. This helps to assess the significance of correlations between features with permutation tests. While a general overview of the correlations between different features and forcings is given in Appendix D, the most important results of our statistical analysis are presented in the following Sections.

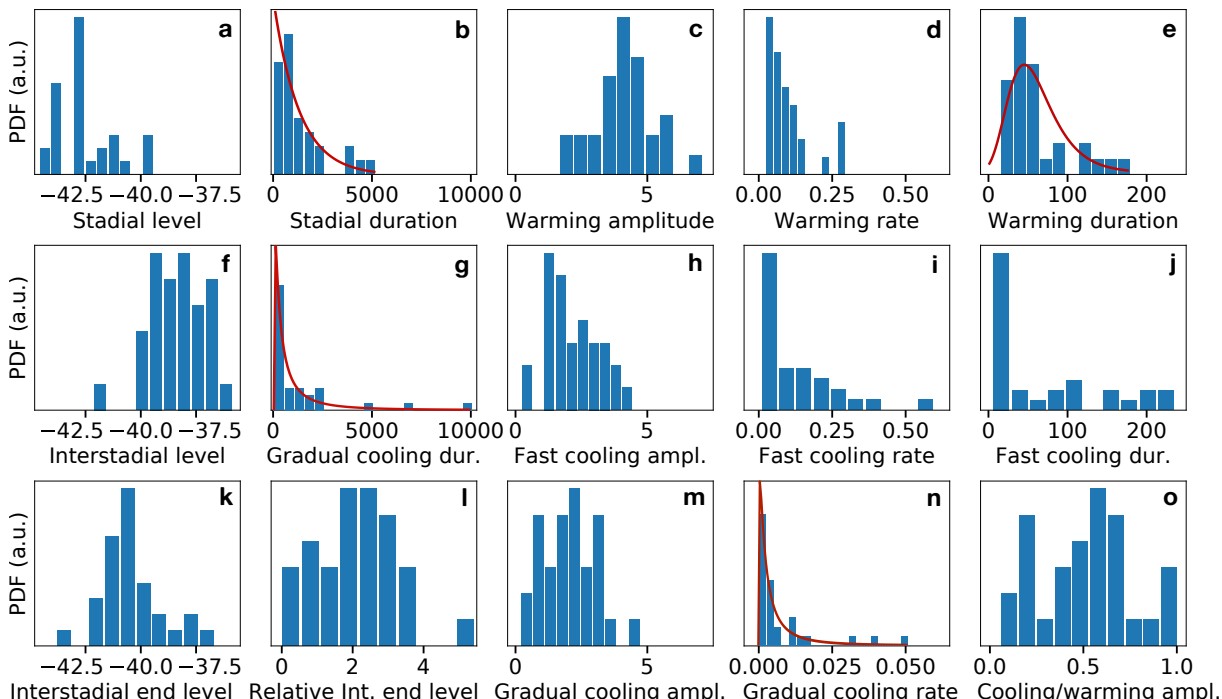

**Figure 4.** Histograms of our sample of 31 events for all features considered in this study, as defined in Tab. 1. The *red* curves in panels b) and e) are fits with the exponential and Gumbel distributions, respectively, whereas those in panels g) and n) are fits with the log-normal distribution.

## 4.1    Interstadial periods

### 4.1.1    Cooling rates determine interstadial durations

10    We focus on the factors influencing the durations of the interstadial periods, defined as $b_3 - b_2$. As noted previously for the younger half of the last glacial in the GISP2 ice core, there is a correlation of interstadial durations $D_I$ and their respective cooling rates $\lambda_c = A D_I^{-1}$, where A is the amplitude of the cooling (Schulz, 2002). If for every interstadial the gradual cooling





would be perfectly linear and the jump back to stadial conditions would always occur after the same magnitude of cooling $\bar{A}$, the interstadial duration would be inversely proportional to the cooling rate $D_I = \bar{A}\lambda_c^{-1}$. Conversely, if the interstadials would have a fixed cooling rate $\bar{\lambda}_c$ and the jump back to stadial conditions happened after variable cooling magnitudes, the interstadial durations would be proportional to the cooling amplitudes $D_I = A\bar{\lambda}_c^{-1}$.

We test which of the two scenarios is better supported by the data. This depends on whether either the cooling amplitudes or the cooling rates have a larger spread than the other. The coefficient of variation for the amplitudes is $CV = 0.51$, whereas for the rates we find $CV = 1.49$. The Spearman correlation of durations and cooling rates is $r_s = -0.89$, which is clearly significant given the samples size of 31 events and weak autocorrelation of the sequence of interstadial durations and rates. On the contrary, for durations and cooling amplitudes we find $r_s = 0.40$, which is mainly due to two outliers, the two longest inter-

stadials GI-23.1 and GI-21.1. Removing these reduces the correlation to $r_s = 0.30$, which is not significant at 95% confidence. As a result, there is no relationship between durations and amplitudes that goes beyond outlier events, as opposed to durations and cooling rates. Furthermore, the correlation of cooling amplitudes and rates is not significant. Thus there is indeed a strong control by the cooling rates on the interstadial durations over the entire glacial, in agreement with the findings for the younger half of the glacial by Schulz (2002).

In Fig. 5a we show a scatterplot of $\log \lambda_c$ and $\log D_I$ along with a linear regression yielding a slope of -0.94. The 95% confidence interval of this slope obtained via bootstrapping is [-1.12, -0.75]. Because we do not account for errors in the rates estimated from the data the regressed slope is biased towards 0 due to attenuation and the true slope will be closer to -1. The model $D_I \propto \lambda_c^{-1}$ is consistent with the data, where the spread is caused by the fact that the jump back to stadial conditions happens after varying cooling amplitudes, which have a mean of 2.04 and standard deviation of 1.04.

**4.1.2  Distribution of interstadial cooling rates and durations**

The relationship between interstadial durations and cooling rates also manifests itself in the respective distributions. As seen in Fig.'s 4g and 4n, both features have strongly skewed histograms. Both are consistent with log-normal distributions, as shown by Anderson-Darling tests with $p = 0.47$ and $p = 0.89$ for durations and rates, respectively. A fit with this distribution is illustrated in the figure. Because the two features are inversely related with $D_I = \bar{A} \cdot \lambda_c^{-1}$, the fact that one is a log-normal

random variable implies that the other is, too. If $D_I$ is distributed log-normally with parameters $\mu$ and $\sigma$, then $\lambda_c^{-1}$ follows a log-normal distribution with parameters $-\mu + \ln(\bar{A})$ and $\sigma$. In our data this relation holds: We estimate $\mu$ and $\sigma$ from the data $D_I$ and use the observed average amplitude $\bar{A} = 2.04$. An Anderson-Darling test with $p = 0.33$ shows that the data $\lambda_c^{-1}$ is consistent with a log-normal distribution with $-\mu + \ln(2)$ and $\sigma$.

As opposed to other skewed distributions like the exponential, Gumbel and power law, both durations and cooling rates are

also consistent with an inverse Gaussian distribution. The observation that the durations and rates and are both well fitted by the inverse Gaussian despite their inverse relation is explained by the similar shape of the reciprocal inverse Gaussian distribution. If a variable is distributed as inverse Gaussian $X \sim IG(x)$, then the distribution of $Y = \frac{\bar{A}}{X}$ is reciprocal inverse Gaussian $Y \sim \frac{\bar{A}}{x^2} IG(\bar{A}/x)$. A moderately sized sample of $Y$ is still likely to be consistent with an inverse Gaussian distribution, due to the similarity of the two. The inverse Gaussian could make an appealing model for the interstadial durations, since it arises as





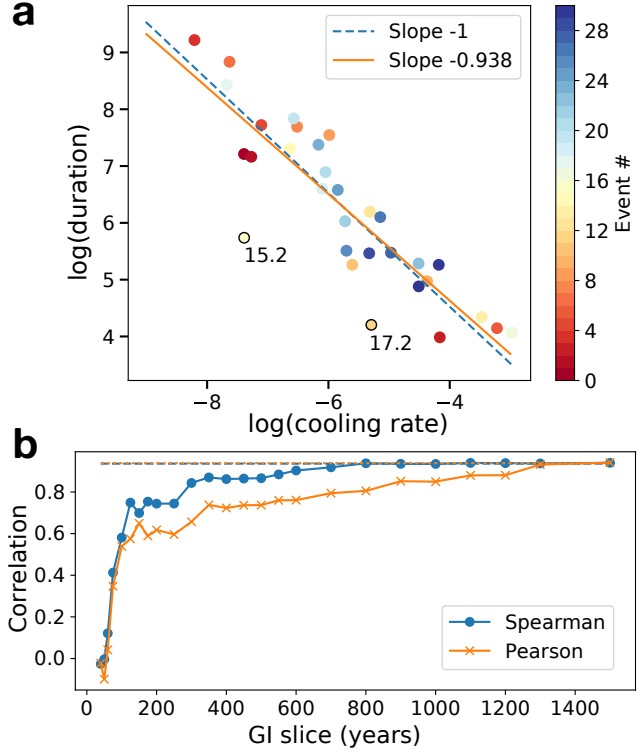

**Figure 5. a)** Scatterplot of the logarithms of interstadial durations and cooling rates. The color coding indicates the temporal sequence of the events starting with GI-24.2 as event #0. Two linear fits obtained by ordinary least squares are shown. For one of them we fixed the slope to -1 and varied only the intercept. **b)** Correlation coefficients of the logarithms of interstadial duration and the linear slope fitted to a slice of the beginning of the interstadial as a function of the length of that slice. The values of the Spearman (Pearson) correlation coefficients using slopes obtained from the full interstadials is marked with a *dashed* (*dotted*) line.

distribution of first hitting times of a constant level for Brownian motion with a constant drift. However, the proxy time series in interstadials look qualitatively different than what is expected from this model, because they are quite linear and yet have strongly varying slopes. In order for the model to produce roughly linear time series, the drift has to be high, which results in very similar slopes of the time series with the resulting distribution of first hitting times converging to a Gaussian. We leave a

5  further discussion on which mechanism could yield log-normal or inverse Gaussian distributions of durations or cooling rates for upcoming studies. Instead, in the following we focus on implications of the approximate linearity of the interstadial time series.

### 4.1.3  Predictability of interstadial durations

The simple relationship of interstadial durations and cooling rates might have some implications on the understanding of DO

10  event dynamics. If the rates imply the durations, then the durations are already determined as soon as the rate is established,





which might happen early in the interstadial. This is different from the idea that the transition from interstadial to stadial might be a noise-induced escape from one metastable state to another. To test this, we take small slices of the beginnings of each interstadial, fit a linear slope $s$ to them and then calculate how well these slopes determine the durations of the full interstadials as we increase the length of the slices. Due to noise in the beginning of the interstadials, for some interstadials a small positive

slope is being detected. We reset these slopes instead to $s = -0.0001$, because in our analysis we use the logarithms of slopes and durations. For the relatively short events 15.2 and 17.2, no negative slopes are obtained when fitting the whole interstadial part independently, as opposed to the slopes obtained in the fit of the entire time series. We thus have to exclude these two outliers in the following. In Fig. 5b we show how the correlation between the logarithm of the slopes $-\log(-s)$ of these slices and the durations $\log D_I$ evolves as we increase the length of the slices. The correlation of the slopes estimated from the full

interstadials and the durations, when excluding events 15.2 and 17.2, is $r_s = 0.94$ ($r_p = 0.94$), and is indicated by a dashed (dotted) line. We can see that the correlation rapidly increases up to a length of 150 years. Thereafter the correlation stabilizes until another more rapid increase at a length of 350 years. The rapid increase in correlation is partly due to a non-negligible number of events already being at full length (6 events at 150 years and 12 events at 350 years). Still, also the slopes of the remaining events already correlate well with the durations. At 350 years, the durations are almost as well determined by the

slopes estimated from the slices as they are from the full interstadials. The remaining indeterminacy comes from a handful of longer interstadials (23.2, 22, 14 and 11) that do not settle to a clear negative slope after 350 years. For the latter three events, this is due to sub-events that occur shortly after the interstadial onset. Although we can see that there are exceptions, we conclude that for most events the interstadial duration is determined at a relatively early stage within a few hundred years after the rapid warming.

### 4.1.4 Influence of external forcing

Having established control of the interstadial durations by the cooling rates, we investigate whether the variability in the rates can be explained by other features of the DO cycles, or by external forcing. Among correlations of the cooling rates with other features deemed significant by a permutation test, none of them are relevant, either because they are caused by few outliers or else directly due to their definition and parameter constraints.

Considering external climate factors, we find a correlation of $r_s = 0.40$ and $r_p = 0.35$ of the logarithm of the cooling rates with LR04 at the time of the interstadial maximum. This correlation is, however, influenced by a common trend of the two quantities, and is not significant anymore at 90% confidence when removing a linear trend. On the other hand, there is a large sub-set of events which appears to be linearly related.

As shown in Fig. 6a and c, the furthest outliers from an approximate linear relationship are the interstadials 23.2, 21.2,

16.2 and 15.1. When removing these outliers, the correlation is $r_p = 0.79$, which clearly goes beyond a common trend with $r_p = 0.63$ after linearly detrending. When only considering the younger half of the record starting with GI-14, the correlation is $r_p = 0.84$. This corresponds to the finding by Schulz (2002), who reports that the interstadial cooling rates starting from GI-14 are forced by global sea level. We note, however, that the correlation after GI-14 is largely due to common trend, as we find





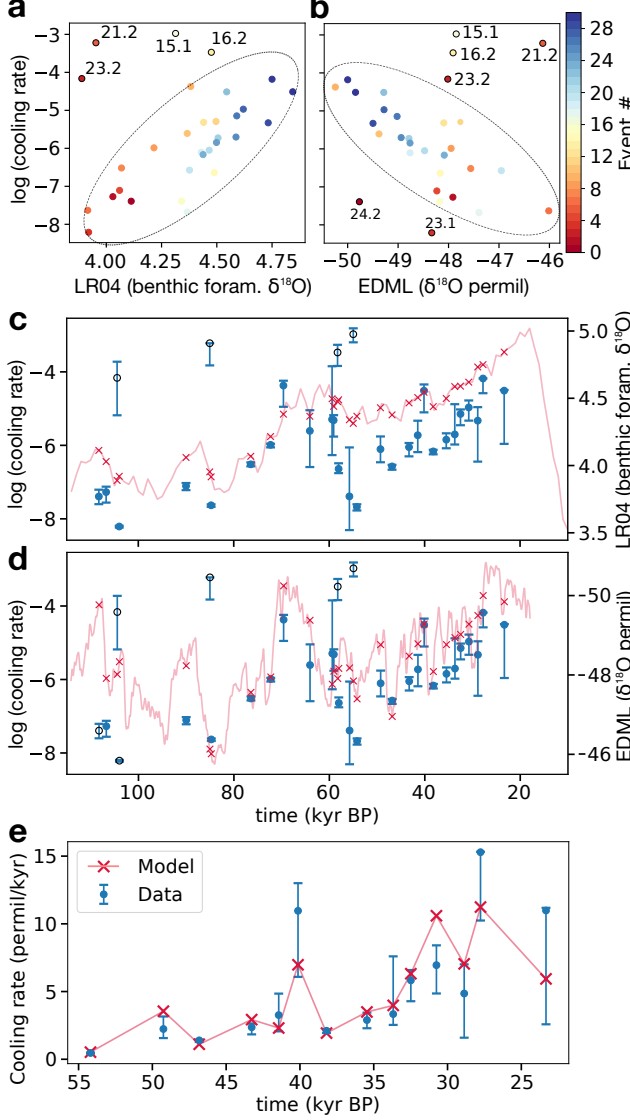

**Figure 6. a-b)** Scatterplot of the logarithm of the interstadial cooling rates and the **b)** LR04 and **b)** EMDL at time points corresponding to the interstadial onsets. **c)** Time series of the cooling rates (*dots*) and the LR04 stack (*crosses*). The error bars on the cooling rates are given by the 16- to 84-percentile obtained by bootstrapping. **d)** Time series of the cooling rates (*dots*) and the EDML stack (*crosses*). Note the inverted axis for EDML. **e)** Time series of the interstadial cooling rates starting at GI-14 and of a linear regression model of the $CO_2$ record fitted to the logarithm of the cooling rates.

$r_p = 0.37$ after linear detrending, which is not significant at 90% confidence. Nevertheless, as shown above, when discarding a few outliers there is evidence for significant correlation as we include older parts of the record.



A better predictor of the interstadial cooling rates of the more recent DO cycles is given by the $CO_2$ composite record. Whereas for the older half of the glacial there is no significant correlation, when starting at GI-14, we find a linear relationship with $r_p = -0.93$ and $r_p = -0.81$ after linear detrending. In Fig. 6e we illustrate how well the cooling rates of this period can be predicted from $CO_2$ by linearly regressing $CO_2$ onto the logarithm of the cooling rates and then exponentiating the result.

Additionally, in a subset of the events, there is a linear relationship between the logarithm of the cooling rates and EDML at the interstadial onsets. While the correlation of the entire data set is not significant at 90% confidence with $r_p = -0.19$ and $r_s = -0.23$, when removing events 24.2, 23.2, 23.1, 21.2, 16.2 and 15.1, the remaining events appear to have an approximate linear relationship, as indicated in Fig. 6b and d. The correlation then becomes $r_p = -0.81$ and $r_s = -0.78$, or $r_p = -0.72$ and $r_s = -0.61$ after linearly detrending, which is significant at 99% confidence. Thus, in this subset there is evidence for

anti-correlation beyond a simple linear trend. Again, the linear relationship is strongest for the younger half of the record, which starts at GI-14 and does not have outliers. Here, we find $r_p = -0.89$, and $r_p = -0.70$ after linearly detrending, which is significant at 99% confidence.

A corresponding linear relationship between the logarithms of interstadial durations and Antarctic temperature in different ice cores has been noted before by Buizert and Schmittner (2015). In our data we find correlations of these quantities of

$r_p = 0.29$ and $r_s = 0.27$ which are not significant at 90% confidence. This disagreement comes from the fact that Buizert and Schmittner (2015) lump each of the interstadials 24, 23, 21, 17, 16, 15 and 2 together into one event, even though they are comprised of two events. If we remove the events 24.2, 23.2, 23.1, 21.2, 17.2, 16.2 and 15.1, we find a strong linear relationship of $r_p = 0.91$, comparable to the findings by Buizert and Schmittner (2015). It is robust to linear detrending with $r_p = 0.87$. Most of these outliers are very short events, and discarding them removes a lot of the variability of the interstadial durations,

similar to lumping them together with adjacent longer events.

## 4.2  Stadial periods

### 4.2.1  Stadial duration distribution

The stadial periods are defined to start after the rapid cooling and end at the onset of the rapid warming, and their duration is thus $b_1$. These durations are highly variable, ranging from 20 years for GS-24.2 to 5169 years for GS-19.1, with an average

of 1328 years. Due to our definition of stadials GS-24.2 is exceptionally short, because the proxy shows rapid warming again right after the rapid cooling without stabilizing. Figure 4b shows that the stadial duration distribution is skewed. The data is consistent with an exponential ($p = 0.79$ with Anderson-Darling test) and a log-normal distribution ($p = 0.18$). A relative likelihood test prefers the exponential distribution by a factor of 16 over the log-normal. An exponential fit to the data is illustrated in Fig. 4b. This distribution might be relevant in this context, as it arises in the low noise limit of noise-induced

escape times from asymptotically stable equilibria in dynamical systems (Day, 1987).





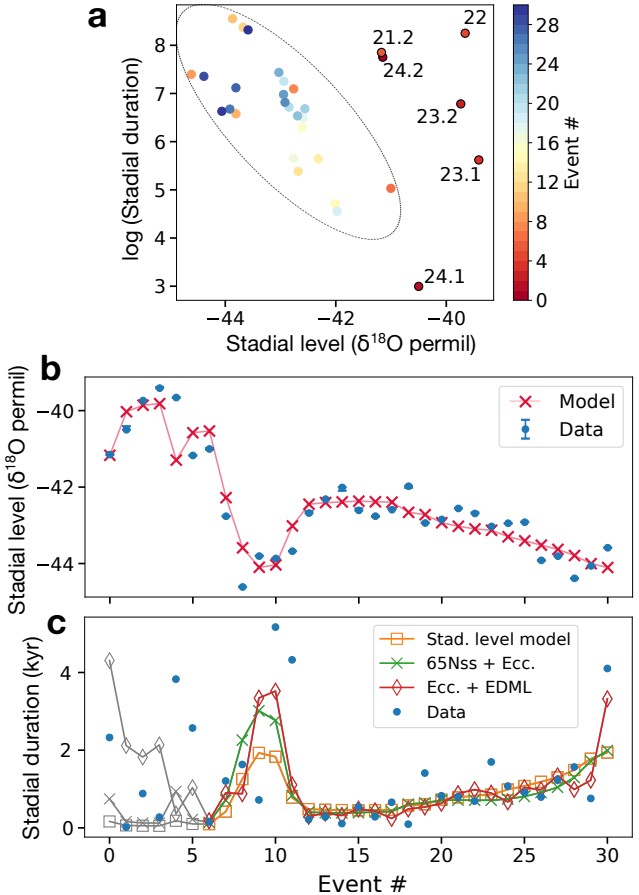

**Figure 7. a)** Scatterplot of stadial levels and logarithmic durations. Outliers from an approximate linear relationship are labeled. **b)** Event series of observed stadial levels and those modeled by $L_{mod} = 3.52 \cdot X_1 + 98.84 \cdot X_2 - 57.96$, where $X_1$ is 65Nint and $X_2$ the eccentricity. **c)** Models predicting the observed stadial durations (*crosses*). The first 6 events, indicated by *gray* markers, were discarded when fitting the models. The model based on predicted stadial levels from insolation (*squares*) is given by $\log(D_{mod}) = -0.90 \cdot L_{mod} - 32.18$. The second model (*circles*) is given by $\log(D_{mod}) = -0.037 \cdot X_1 - 27.11 \cdot X_2 + 25.24$, where $X_1$ is 65Nss and $X_2$ eccentricity. The third model (*diamonds*) is given by $\log(D_{mod}) = -0.90 \cdot X_1 + 75.39 \cdot X_2 + 38.71$, where $X_1$ is EDML and $X_2$ eccentricity.

### 4.2.2 Influence of stadial levels and forcing on durations

In the following we discuss whether the stadial duration variability is influenced by other features in the data, or external factors. Among external factors, the stadial durations are best correlated with 65Nss ($r_s = -0.64$). The only DO feature that is significantly and robustly correlated with the stadial durations, is the stadial levels with $r_s = -0.43$. In Fig. 7a we show a

5    scatterplot of the stadial levels and the logarithms of the durations. If one discards the first 6 events of the record, there is a linear anti-correlation of $r_p = -0.80$, or $r_p = -0.76$ after linear detrending. This could be either due to common forcing or due to the levels directly controlling the durations. While the stadial levels correlate well with LR04 and EDML due to a common linear



trend, there is better correlation with insolation, as seen by $r_p = 0.60$ (GS-24.2 and GS-22 are outliers) for 65Nss. Removing outliers yields $r_p = 0.82$, which does not change when linearly detrending. To see whether this forcing explains most of the correlation of durations and levels, we remove a linear fit to 65Nss from each variable and find a remaining correlation of $r_p = -0.38$. Even though the significance of this correlation is unclear due to the autocorrelation of the stadial levels, this could imply that there is more information in the stadial levels about the durations than simply common insolation forcing. On the other hand, insolation components in addition to 65Nss might explain more of the observed variability.

We investigate whether multiple linear regression models with two predictors explain the stadial levels and durations significantly better. A model comprised of 65Nint and eccentricity determines the stadial levels very well ($R^2 = 0.86$), as shown in Fig. 7b. These modeled levels also correlate well with the logarithm of the stadial durations ($r_p = -0.64$ when excluding the earliest 6 events). We check whether this is a good model for the durations by linearly regressing the modeled levels onto the logarithm of the durations and exponentiating the result. In Fig. 7c we compare this to two other models that directly regress external forcings on the logarithm of the durations. None of the models fits the first 6 events adequately. Thereafter, all three models produce a similar trend. The model based on predicted stadial levels, and a model with direct forcing by 65Nss and eccentricity show similar skill with $R^2 = 0.29$ and $R^2 = 0.30$, respectively. The third model based on eccentricity and the EDML record is slightly better with $R^2 = 0.46$, mainly because it fits two of the longest stadials better. Still, all of the models fit only the overall trend and leave unexplained variability on top of it. Unless the correlation is nearly perfect, a linear correlation of the logarithm still leaves a lot of room for scatter in the original scale.

The exponential tail in the variability of the stadial durations is not a result of the modulation by the external forcings we consider. To demonstrate this, we remove the forcing influence by fitting a linear model of one or more forcings to the logarithm of the stadial durations. We obtain detrended data by adding the mean of the logarithmic data to the residuals of the fit and then exponentiating. When using 65Nss as forcing, we find $p = 0.15$ in an Anderson-Darling test on the exponential distribution. With the model of both eccentricity and 65Nss, as introduced above, we find $p = 0.29$. Thus, the distribution of the detrended data is still long-tailed and consistent with an exponential distribution.

## 4.3 Abrupt warming periods

### 4.3.1 Warming durations

The rapid warming transitions in NGRIP as determined by our piecewise-linear fit have an average duration of 63.2 years. There is quite a large spread with a minimum duration of 15.3 years for GI-17.1 and a maximum of 179.5 years for GI-11. There is no clear trend, as we find both short and long warmings in early and later parts of the record. The distribution is skewed as seen in Fig. 4e. We find 5 transitions that last for more than a hundred years (interstadials 6, 11, 17.2, 18, 23.2). For each of them there is not only a single abrupt warming, but also a systematic departure from stadial to warmer values before, as can be seen in Fig. S1 of the supplemental material. Our algorithm includes these early warming trends into the warming transition. Clearly, other methods to define the abrupt warmings might give different results in these cases. In Rousseau et al. (2017), the transition onsets are defined by the derivative of the signal and consequently the warming transitions into interstadials 6 and



11 are reported to be much shorter. Given our definition of abrupt warmings, we can at least argue that the longest warming transitions are not a result of noise, because in our fit of the GRIP record the same transitions are also among the longest and are clearly above average.

Within the framework of our analysis, we cannot identify any DO cycle features, external forcings, or combinations thereof
that can explain a significant part of the variability in the warming durations. Thus, we aim to infer something about the mechanism of the warming transitions from the distribution of their durations. To assess which distributions are consistent with the data we use the Anderson-Darling test. Using the Cramer-von Mises or Kolmogorov-Smirnov test yields qualitatively unchanged results for all tests reported in this section. The Anderson-Darling test shows that the log-normal (p=0.63), Gumbel (p=0.053) and inverse Gaussian (p=0.95) distributions cannot be rejected at 95% confidence by the data. A fit with the Gumbel
distribution is illustrated in Fig. 4e. By computing the relative likelihood from the Akaike information criterion, we find that the inverse Gaussian distribution is 9.7 times more likely than the Gumbel distribution, and the log-normal distribution is 7.6 times more likely than the Gumbel distribution. We cannot choose in between log-normal and inverse Gaussian with any confidence.

### 4.3.2 A model for the stadial-interstadial transition

In the following we compare the warming durations to what is expected in the framework of noise-induced transitions in
multi-stable systems. The DO warming durations are much shorter than the time spent in the stadial state. If we consider the stadial-interstadial transition as a noise-induced transition from one metastable state to another, starting at the stadial onset, most of the time is spent in the vicinity of the stadial state. The part of the trajectory that leaves this vicinity for the last time and then moves towards the other state (interstadial) is referred to as the reactive trajectory. Because of the high noise level in the record, an unknown part of which is non-climatic or regional and changes over time, we do not estimate reactive trajectories
by defining neighborhoods of two metastable states. Instead, we believe the warming periods obtained by our piecewise-linear fit are reasonable estimates. Figure 8a illustrates the reactive trajectory (in green) leading up to GI-20. In Fig. 8b the different parts of this stadial-interstadial transition are projected onto an arbitrary potential that features two metastable states. For overdamped motion driven by additive noise in such a potential, it has been proven that the reactive trajectory durations converge to a Gumbel distribution in the zero noise limit (Cérou et al., 2013). Similarly, there is numerical evidence for the
Gumbel distribution applying to one-dimensional spatially extended systems for low noise (Rolland et al., 2016). Because in our data we cannot separate true climatic noise, potentially driving the observed large-scale climate transitions, from other types of noise, it is hard to say whether a low noise condition is met and a Gumbel distribution should be expected.

With a small numerical experiment we address the case of finite noise levels and small sample sizes. We use stochastic motion in a double well potential as generic model for a noise-induced transition out of a metastable state to another. It is
given by the stochastic differential equation $dX_t = \left(-\frac{dV(X_t)}{dx}\right)dt + \sigma dW_t$, with the potential $V(x) = x^4 - x^2$ and the Wiener process $W_t$. For zero noise, there are two fixed points at $x = -1$ and $x = 1$. We initialize the system at $x = -1$ and repeatedly collect reactive trajectories, which start when they last leave $x < -0.9$ and end as they enter $x > 0.9$. Small samples of 31 reactive trajectory durations are indeed typically consistent with a Gumbel distribution for a range of different noise levels, but can be consistent with other distributions, too.





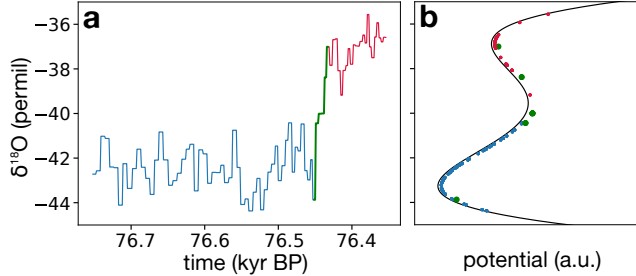

**Figure 8. a)** Stadial-interstadial transition leading up to GI-20 (*red*) including our estimate of the so-called reactive part of the trajectory (*green*) preceded by 350 years of the stadial GS-21.1. **b)** Data points of the same time series projected onto an arbitrary one-dimensional potential function with two minima as conceptual model for the transition.

To show this, we collect p-values of Anderson-Darling tests on many small samples. For the Gumbel distribution at a low noise level of $\sigma = 0.00045$, 96.3% of the p-values are above 0.05. Thus, in this case, very rarely a sample of 31 reactive trajectory durations is rejected by a hypothesis test on the Gumbel distribution. For a higher noise level of $\sigma = 0.5$, still 80.1% yield $p > 0.05$. However, the log-normal distribution fits equally well with 95.4% (93.6%) yielding $p > 0.05$ for $\sigma = 0.00045$

($\sigma = 0.5$). The distribution that most reliably fits the data is the inverse Gaussian with >99.9% (>99.9%) yielding $p > 0.05$ for $\sigma = 0.00045$ ($\sigma = 0.5$), despite the fact that in the zero noise limit the correct distribution is Gumbel. It has been noted that the inverse Gaussian also fits well for large sample sizes (Cérou et al., 2011). Even non-skewed distribution can be consistent with the samples, as seen for the Gaussian distribution, with 55.1% (22.8%) yielding $p > 0.05$ for $\sigma = 0.00045$ ($\sigma = 0.5$). Similar values are obtained for the logistic distribution.

These results imply that from a small sample of 31 reactive trajectories we cannot reliably identify the true distribution and thus a potential mechanism. Still, the data are at least consistent with the expected behavior of noise-induced escape from a metastable state. Clearly, other simple mechanisms can be consistent with the data, too. For example, as mentioned above, the inverse Gaussian is the distribution of time elapsed for a Brownian motion with drift to reach a fixed level.

### 4.3.3 Warming amplitudes

The average amplitude of the rapid warmings is 4.2 permil, with most events clustering around this value. The most extreme values are 7.1 permil for GI-19.2 and 1.7 permil for GI-5.1. The latter is not surprising, because GI-5.1 is almost not visually discernible as an event in the $\delta^{18}O$ series. We find an anti-correlation of the warming amplitudes and the preceding stadial levels. When discarding GI-5.1, the correlation is significant at 99% confidence with $r_s = -0.63$, which is largely due to GI-19.2 being preceded by a very deep stadial, and GS-23.1 and GS-22, which are preceded by very shallow stadials, as they

happen early in the glacial. When discarding these events the remaining correlation is still significant at 99% confidence with $r_s = 0.50$. Thus, to some degree, the warming amplitudes are predictable in a statistical sense: When residing in a shallow stadial, the amplitude of the next DO warming will be small, and vice versa for a deep stadial. We also assess whether the variability can be explained by external forcing. Our analysis does not show a relationship between the DO event amplitudes





and global ice volume (LR04), as has been proposed by McManus et al. (1999); Schulz et al. (1999). It should be noted, however, that these studies have a quite different notion of DO event amplitudes. Our approach, based on fitting high resolution data, seems well suited to estimate the actual amplitude of rapid transitions, as opposed to low-pass filtering that reduces the amplitude of shorter events. Instead of ice volume, we find a correlation with 65Nint of $r_p = -0.36$ and $r_s = -0.31$, which is

significant at 90% confidence. However, the correlation is visually not striking. Removing GI-19.2, which occurs close to an insolation minimum, yields a correlation that is not significant at 90% confidence.

## 5 Discussion

This work presents a consistent fitting routine allowing to extract robust features of DO events from noisy, high-resolution ice core data, such as the NGRIP $\delta^{18}$O record. The algorithm converges to a continuous piecewise-linear fit of the whole

time series, where each DO event cycle is given by a constant stadial period, an abrupt warming period, a gradually cooling interstadial period and an abrupt cooling period. The fit is satisfactory in the sense that each event receives a reasonable saw-tooth shaped fit, which is characteristic for the ensemble of DO events as a whole. Not for all events this is necessarily the overall best piecewise-linear fit. For example, there are transitions that do not have a significant rapid cooling period at the end of the interstadials, but rather cool gradually until reaching roughly constant stadial values. In Sec. 3.1 we showed that 14 out

of the 31 DO events analyzed in this study do not strictly follow the saw-tooth shape that is often reported as being generic for all DO events.

The uncertainties of the fit parameters are assessed by using a bootstrap resampling technique (Sec. 3.2) and alternatively by comparison to a different ice core (GRIP, Sec. 3.3). The average absolute deviations of the GRIP and NGRIP features and the average standard deviations obtained by bootstrapping are of very similar magnitude. This gives us confidence in the validity

of the uncertainty estimates of the latter method. From the uncertainties it follows that some of the shorter time scale features have to be taken with care, such as the rapid warming durations. Here, not all individual values might be reliable. However, the comparison with GRIP shows that the overall trends and distributions, also of the shorter time scale features, are robust. Still, different methods or models to define the features might alter the results. As an example, our piecewise-linear method yields quite different estimates of the abrupt warming durations as compared to Rousseau et al. (2017), where abrupt warmings

are defined by an estimated derivative of the time series. Our results have an average absolute deviation of 25 years (26 years) compared to their algorithmically (visually) determined warming durations starting at GI-17.1.

We subsequently analyzed different features that describe each DO cycle and can be derived from the fit parameters. These include the proxy levels in the stadials, at the interstadial maxima and ends, the durations and rates of warming and cooling periods, as well as stadial and interstadial durations. In general, all features except for the proxy levels develop rather irregularly

from event to event, as shown by the absence of significant autocorrelation, and many of them are very broadly distributed. We evaluated which distributions and corresponding processes describe the individual features best. This can give some insight into the nature of the mechanisms giving rise to the abrupt climate changes. Furthermore, we investigated whether the variability in some features can be explained by other features, external forcings or combinations of them. This is done by a brute force





search of significant correlations, multiple linear regression models and clusterings, which is subsequently narrowed down by assessing robustness to outliers and trends. We synthesized the most important findings in terms of the interstadial, stadial, and abrupt warming periods of the DO cycle, as summarized in the following.

The interstadial periods have highly variable durations and are characterized by a roughly linear cooling with rates that vary
strongly from one DO cycle to the next (Sec. 4.1.1). The cooling rates clearly determine the interstadial durations, as opposed to the cooling amplitudes, which cannot robustly explain the variability of the durations. Interstadial durations and cooling rates are consistent with a simple inverse relationship. As a result, the interstadial durations are determined to a good approximation as soon as the cooling rates have stabilized. We estimate from the data that for most DO cycles this happens within the first 150 to 350 years of the interstadial (Sec. 4.1.3).

The influence of external factors on the large variability of the interstadial features is assessed and compared to previously proposed forcing mechanisms (Sec. 4.1.4). Based on the GISP2 ice core record of the younger half of the last glacial it has been proposed, that the interstadial cooling rates are controlled by global sea level (Schulz, 2002). While we can confirm this finding by observing a linear correlation of the logarithm of the cooling rates with the LR04 record for global ice volume, the relation seems to be weak in the older half of the glacial due to a handful of outliers. The interstadial duration might be controlled
by influences of global ice volume on the strength and stability of the interstadial (strong) mode of the Atlantic Meridional Overturning Circulation (AMOC). However, contrary to our finding, the influence of ice volume on AMOC stability reported by studies with globally coupled models is often such that increases of Northern Hemisphere ice sheets actually enhance the stability of the strong AMOC state, which would intuitively result in longer interstadials (Zhang et al., 2014). This has been addressed by Buizert and Schmittner (2015), where Southern Ocean processes are invoked to influence interstadial durations.
We find that the control of Antarctic temperature on interstadial durations reported in this study is only valid if certain outliers are discarded. Finally, for the younger half of the glacial, starting at GI-14, we find that the $CO_2$ record is the best predictor of the cooling rates.

In our analysis, certain interstadials frequently show up as outliers. These include the short events 23.2 and 21.2, which occur early in the glacial and are surrounded by longer interstadials, as well as the events 18, 17.2 and 15.2, which are short,
but do not show a clear gradual cooling. Their presence either showcases the strong irregularity and variability of the processes underlying the DO cycle, or could indicate that not all transitions are caused by the same trigger. We find more groups of outliers using clustering analysis. These could be studied more thoroughly, and compared with other paleoclimatic indicators.

The stadials have different properties compared to the interstadials, going beyond the approximately constant temperature within stadials. The duration distribution closely resembles an exponential, and is thus consistent with noise-induced escape
from a metastable state to another (Sec. 4.2.1). The large dispersion of this distribution cannot be explained by external forcing alone. Instead, the distribution is still consistent with an exponential after detrending with the best fit to insolation forcing (Sec. 4.2.2). Although there are different possibilities to obtain a good fit of the trend of the stadial durations, a forcing by insolation seems most plausible. We additionally find indications for a control of the stadial durations by the levels, but it is difficult to conclude from our data whether there is a true causal link, or merely common insolation forcing on both variables.

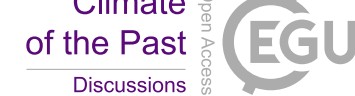

The piecewise-linear fit furthermore gives estimates for the amplitudes, rates and durations of the rapid DO warmings. We find no evidence for influence of other DO features or external factors on the warmings. As an exception, the warming amplitudes are anti-correlated with the stadial levels (Sec. 4.3.3). Consequently, they may be predicted in a probabilistic sense. There is considerable variability of the warming durations, and we find the distribution to be consistent with the durations of

so-called reactive trajectories in systems with noise-induced escapes in between multiple metastable equilibria (Cérou et al., 2013; Rolland et al., 2016) (Sec. 4.3.2).

Thus, our analysis suggests that both the stadial period durations and subsequent warming durations are consistent with the stadial-interstadial transition as a noise-induced escape from a metastable state. This is different from the interstadial-stadial transition, which occurs in a more predictable fashion, because the linear cooling rates anticipate the interstadial durations. Ad-

ditionally, the interstadial durations are not consistent with an exponential distribution. It has, however, recently been suggested that there is a bifurcation before DO events in a fast sub-system of the climate, which was based on evidence for critical slowing down in the high-frequencies of the ice core record prior to a significant number of DO warming transitions (Rypdal, 2016; Boers, 2018). If this is the case it would mean that there is some predictability of the warming transitions, too. Additionally, we find that the influence of external forcing is different for stadial and interstadial periods, with more evidence for insolation

forcing on stadials and ice volume or $CO_2$ on interstadials, which is related to the findings by Lohmann and Ditlevsen (2018). Except for a common forcing envelope of stadial and interstadial levels, there is no strong relationship in between features across the different periods of the DO cycle.

## 6   Conclusion

We developed an iterative method to fit a continuous piecewise-linear waveform to the entire last glacial $\delta^{18}O$ record, which

converges well. By using parameter constraints, we can fit a characteristic saw-tooth shape to every DO event. However, we find that for many of the transitions this is ad-hoc. Almost half of the events do not show a distinct and significant rapid cooling episode after the more gradual interstadial cooling. An analysis of the DO event features derived from the fit confirms the irregularity and randomness that is evident from visual inspection of the record. There is hardly any evidence for relationships linking the features that describe the stadial, interstadial and abrupt warming periods, except for a common envelope that

governs the stadial and interstadial levels via external forcing. A statistical analysis hints at different mechanisms underlying warming and cooling transitions. This manifests itself in different distributions and external influences of the stadials and interstadial durations, as well as the fact that the interstadial durations can be predicted to some degree by the linear cooling rates shortly after interstadial onset.





## Appendix A: Iterative algorithm to fit piecewise-linear model

---

**Algorithm 1** Pseudocode for fitting algorithm

---

1: **while** $j < J$ **do**

2:    **if** $j = 0$ **then**

3:       $\{t_i^b\} = \{t_i^{b0}\}; \{t_i^e\} = \{t_i^{e0}\}$

4:    **else**

5:       Set Stadial durations: $\{D_i^{St}\} = \{b_{1,i} + t_i^b - t_{i-1}^b - b_{4,i-1}\}$

6:       Set Interstadial durations: $\{D_i^{Is}\} = \{b_{4,i} - b_{1,i}\}$

7:       Set Stadial beginning times: $\{t_i^b\} = \left\{\sum_{n=0}^{i-1}(D_n^{St} + D_n^{Is})\right\}$

8:       Set Stadial end times: $\{t_i^e\} = \left\{\sum_{n=0}^{i} D_n^{St} + \sum_{n=0}^{i-1} D_n^{Is}\right\}$

9:    **end if**

10:   Define Stadial levels: $\{l_i^s\} = \left\{\left\langle X_{t_i^b,...,t_i^e}\right\rangle\right\}$

11:   Cut into segments: $\{s_i\} = \left\{X_{t_i^b,...,t_{i+1}^e}\right\}$

12:   **while** $i < N$ **do**

13:      **if** j=0 **then**

14:         Initial conditions: $\theta_i^* = \theta_i^0$

15:      **else**

16:         Initial conditions: $\theta_i^* = \theta_i$

17:      **end if**

18:      Find optimal $\theta_i^{new}$ of segment $s_i$ with $\theta_i^*$, $l_i^s$ and $l_{i+1}^s$

19:      $i = i + 1$

20:   **end while**

21:   Update parameters $\{\theta_i\} = \{\theta_i^{new}\}$

22:   $j = j + 1$

23: **end while**

---

In the following, we detail the optimization procedure to find the best saw-tooth shaped fit for each event, i.e., line 18 of the algorithm above. To determine the 6 parameters at each transition, we minimize the root mean squared deviation of the fit from the time series segment. Due to the high noise level, there are many local minima in this optimization problem. Thus, either a brute-force parameter search on a grid or an advanced algorithm is needed to find a global minimum. We chose an algorithm called basin-hopping, which is described in Olson et al. (2012) and is included in the Scientific Python package scipy.optimize, where it can also be customized. The basic idea of the algorithm is the following: Given a initial coordinates in terms of the parameter vector $\theta_0$, one searches for a local minimum of the goal function $f(\theta)$, e.g., with a Newton, quasi-Newton or other method. The argument to this local minimum $\theta_n$ is then randomly perturbed by a Kernel to yield new coordinates $\theta*$, which





are the starting point of a new local minimization. Next, there is a Metropolis accept or reject step: We accept the argument of the local minimization $\theta_{n+1}$ as new coordinates if the local minimum is deeper than the previous one $f(\theta_{n+1}) < f(\theta_n)$, or else with probability $e^{-(f(\theta_{n+1})-f(\theta_n))/T}$, where $T$ is a parameter relating to the typical difference in depth of adjacent local minima. Now we go back to the perturbation step either with old coordinates $\theta_n$ or, if accepted, with new coordinates $\theta_{n+1}$,

and repeat. The iterative procedure is repeated for a large number of iterations and the result is the argument to the lowest function value found.

Within basin-hopping, one has the freedom of choosing any local minimizer as well as perturbation Kernel. These have to be adapted to our optimization problem. We have several constraints on the parameters that need to be satisfied by the optimization. For instance, we demand that all segments of the fit are present and do not overlap ($b_1 < b_2 < b_3 < b_4$). Other constraints ensure

that the characteristic shape of DO events is fit as good as possible for all events. Among other things, we thus demand the gradual slope to be significantly longer and less steep than the fast cooling transition at the end of an interstadial. An overview of all the constraints we used is given further below. To satisfy them, we chose a multivariate Gaussian perturbation Kernel, which is truncated at the respective parameter constraints. The local minimizer choice requires further consideration. Our goal function landscape is very rough and not differentiable. Thus, methods like gradient descent give very poor results in our

case. A method that does not depend on derivatives and can handle constraints is called Constrained Optimization by Linear Approximation (COBYLA), and we found it to work well in our case.

Two hyper-parameters have to be specified in the basin-hopping algorithm: The variance of the perturbation Kernel, and the parameter $T$ used in the Metropolis criterion. These should both be comparable to typical differences in goal function (temperature) and arguments (perturbation width) of neighboring local minima in the minimization problem. We chose these

parameters empirically by observing how the goal function changes as we slightly change the fit. Although this varies significantly from transition to transition, we determined single values as a compromise for all transitions. For the Kernel variance in the directions of $b_{1,2,3,4}$ we chose a value of 15, and for $s_1$ and $s_2$ we chose 0.004 and 0.0015, respectively.

The following list contains all constraints used in the optimization problem in order to ensure convergence of the algorithm to a fit within the qualitative limits of the desired characteristic waveform. Specifically, constraints 3 and 4 shall guarantee that

there is a distinction in between gradual cooling and rapid cooling at the end of an interstadial. With these constraints we can prevent that our algorithm splits an interstadial in half with two very similar slopes, which can easily happen because there are interstadials which arguably have a rather gradual cooling all the way down to the next stadial with no easily discernable steep cooling at the end. The lower limit of constraint 6 shall help to only fit to the steep part of warming transitions, which might have a slight warming prior to it. The upper limit of constraint 7 is needed in order to force a small negative slope on very short

transitions which otherwise could also be viewed as plateaus.

1. No overlap of segments:
   $b_2 > b_1$, $b_3 > b_2$ and $b_4 > b_3$

2. Gradual slope cannot go below following stadial level $l_{i+1}^s$:
   $s_1(b_2 - b_1) + s_2(b_4 - b_3) > l_{i+1}^s$



3. Gradual slope must be twice as long as steep drop:

$$b_3 - b_2 > 2 \cdot (b_4 - b_3)$$

4. Drop at end of interstadial must be at least twice as steep as gradual slope:

$$2 \cdot s_2 < \frac{s_1(b_2 - b_1) + s_2(b_3 - b_2) - l^s_{i+1} + l^s_i}{b_4 - b_3}$$

5. Stadial period not shorter than 20 years:

$$b_1 > 20, \, b_2 > 20, \, b_3 < (D^{St} + D^{Is} - 20)$$
and $b_4 < (D^{St} + D^{Is} - 20)$

6. Limit steepness of up-slope (permil $y^{-1}$):

$$0.02 < s_1 < 1.5$$

7. Limit steepness of down-slope (permil $y^{-1}$):

$$-0.3 < s_2 < -0.0001$$

For the basin-hopping algorithm we use a multivariate Gaussian Kernel of fixed variance with $\sigma_{b_1} = 15$, $\sigma_{b_2} = 15$, $\sigma_{b_3} = 15$, $\sigma_{b_4} = 15$, $\sigma_{s_1} = 0.004$ and $\sigma_{s_2} = 0.0015$.

**Appendix B:  Convergence of iterative fitting routine**

We repeatedly run our iterative fitting routine and monitor whether the individual parameters converge, so that a consistent fit is obtained in the end. Critical for obtaining a consistent fit is that the stadial levels do not change substantially, as explained in the Methods section. In Fig. B1a we show the evolution over 40 iterations of the incremental deviations of the stadial levels compared to the previous iteration. Most stadial levels converge rapidly so that their increments stay below 0.05 permil. Two short stadials keep fluctuating until around iteration 20 before they converge. Because of the convergence of stadial levels, we

consider our fit to be consistent. Furthermore, the best fit parameters are robust, which can be seen in Fig. B1b. Here, we show the average absolute incremental deviations to the break point parameters at each iteration. After 15 iterations the procedure is stable, with average incremental deviations of roughly 0.4 years for $b_1$ and $b_2$ to 0.5 years for $b_3$ and $b_4$, which result from the stochastic fitting algorithm. Note that these values are already well below the smallest sample spacing of the original unevenly spaced time series.

**Appendix C:  Uncertainty estimation of fitting parameters**

Because of the nature of the data, care has to be taken when generating synthetic data. The properties of the data changes throughout the record and are also quite different in between adjacent stadials and interstadials. Stadials have both a larger variance and a larger effective sample spacing in time than the interstadials. For this reason, synthetic data will be created for each stadial and interstadial period individually. The original data is unevenly spaced, which would provide difficulties on its



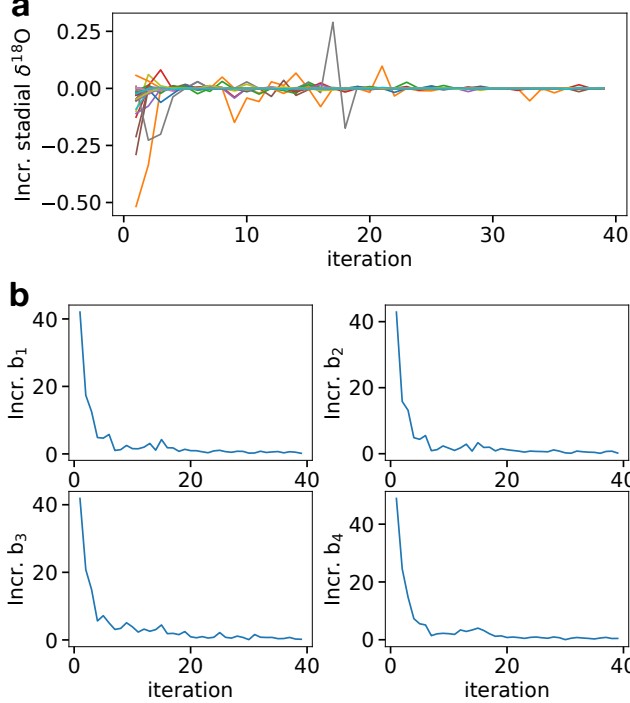

**Figure B1. a)** Evolution of the incremental change of all stadial levels compared to the previous iteration for all 40 iterations of the fitting routine. **b)** Average over all transitions of the incremental change (absolute value) of the break point parameters $b_1$, $b_2$, $b_3$ and $b_4$.

own, while our data is nearest-neighbor interpolated and oversampled to a 1-year resolution. This means that there typically are multiple neighboring point with the same value, making it challenging to find a valid autoregressive or ARMA model for the residuals to generate synthetic data. Instead, we use a block bootstrap resampling technique to keep all relevant structure in the data. We chose a simple block bootstrap where non-overlapping blocks of fixed length of the time series are randomly

5 ordered, because it preserves the correct mean of the individual stadial and interstadial residuals. More involved methods, such as the stationary bootstrap, could be applied, but it likely will not change any of our conclusions.

In the following, we present the procedure for uncertainty estimation. We denote the original data time series of a given transition as $\{X_t\}$, the fit obtained by the data as $\{Y_t\}$ and the residuals to the fit as $\{R_t\} = \{X_t - Y_t\}$. We furthermore use the break points $b_{1,2,3,4}$ obtained in the fit of this transition.

10  1. Divide the residuals into four segment $R_t^i$ at the breakpoints:

   $\{R_t^i\} = \{R_t\}_{t=b_{i-1}...b_i}$ for $i = 1...4$, where $b_0 = 0$.

   Denote the length of $\{R_t^i\}$ as $n_i$.

2. For each segment: Divide into $n_i/l$ blocks of length $l$.

   Append remaining data points to last block if $n_i/l$ non-integer.

15  The block length $l$ is determined by the length of the segment, as explained below.



3. For each segment: Randomly sample blocks without replacement and concatenate until all blocks have been used. This yields resampled segments $\{\bar{R}_t^i\}$.

4. Concatenate the four resampled segments and add the fit to get synthetic data

    $\{X_t^*\} = \{Y_t\} + \{\{\bar{R}_t^1\}, \{\bar{R}_t^2\}, \{\bar{R}_t^3\}, \{\bar{R}_t^4\}\}$

5. Fit $\{X_t^*\}$ to a piecewise-linear model with the basin-hopping algorithm.

6. Repeat from step 2.

In order to also be able to resample the shortest segments, while also preserving the autocorrelative structure in all but the shortest segments, we choose the following scheme for the block length $l$: If the segment length $n_i$ is larger than 40 years, choose $l = 20$. If $40 > n_i \geq 20$ choose $l = 10$. If $20 > n_i \geq 10$ choose $l = 5$. If $n_i < 10$ do not resample and simply return

original segment. The scheme has been determined by looking at the residuals of each segments in all transitions and observing that the autocorrelation drops to non-significant values for all segments after 10-15 years. It thus seems reasonable to use the same block length rule for all transitions and segments.

## Appendix D:  Correlation analysis of features and forcings

In the following, we give an overview of the pairwise correlations in between different features and forcings. We show the

Spearman correlation coefficients of all tests and their significance in Fig. S4 of the Supplementary Material. Considering Spearman correlations with $p < 0.05$, we find 81 positives at 95% and 50 positives at 99% confidence, which is clearly more than expected by chance. However, as detailed in the Methods Section, many of these are due to construction and will not be discussed here. We will furthermore omit correlations which are not robust due to the presence of outliers.

Among features within the same DO cycle, the three different levels yield a strong correlation with each other. However,

the significance is overestimated due to their autocorrelation, and after linear detrending, the correlations are not significant anymore. Thus, the correlation comes mostly from a common trend associated with evolution of the background climate state during the glacial. Furthermore, we find significant correlations of fast cooling, gradual cooling and warming amplitudes, and a correlation of interstadial levels and gradual cooling amplitudes. This implies a certain consistency of DO cycles, where a large amplitude warming is typically also followed by a large amplitude cooling (gradual and/or fast). This is equivalent to the

fact that the stadial levels are autocorrelated. In Sec. 4 we furthermore discuss the correlation of the gradual cooling durations with the gradual cooling amplitudes and rates, as well as the correlation of the stadials levels with the stadial durations and warming amplitudes.

For features in adjacent DO cycles, we do not expect any true positives a priori, because no features are related by construction. Significant correlations at 99% confidence are only found for the levels. Due to their autocorrelation, the significance

determined by permutation tests are not reliable, however. Detrending shows that the correlations are dominated by a common linear trend due to the slowly changing background climate state. The remaining 8 correlations significant at 95% confidence





could either be false positives, or a result of common external forcing. This is because 7 of the 8 correlations involve the levels, which are clearly influenced by forcing, as detailed below.

We furthermore correlate the features with all forcings at the onset times of the respective periods within the DO cycles. The tests indicate clearly more significant correlations than expected by chance. However, due to autocorrelation, the significance is overestimated by permutation tests. In particular, the levels yield significant correlation with most forcings, however, both are autocorrelated. By linearly detrending and discarding outliers where necessary, we find that the interstadial levels are best correlated with LR04, EDML and $CO_2$, the interstadial end levels with 65Nss and precession, and the stadial levels with LR04, 65Nint, 65Nss, obliquity and eccentricity. Additional significant correlations we found are discussed in Sec. 4 and include those of gradual cooling rates with the LR04 and $CO_2$ forcings, as well as those of stadial durations and different insolation forcings.

*Competing interests.* The authors declare no competing interests.

*Acknowledgements.* We gratefully acknowledge discussions of this work with Sune O. Rasmussen. This project has received funding from the European Union's Horizon 2020 research and innovation Programme under the Marie Sklodowska-Curie grant agreement No 643073.





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
