# Peer review of "Objective extraction and analysis of statistical features of Dansgaard-Oeschger events"

_Climate of the Past, 2019_

## Referee Comment (RC1) · Anonymous Referee #1 · 16 Apr 2019

Review of paper "Objective extraction and analysis of statistical features of Dansgaard-Oeschger events" by J. Lohmann and P. Ditlevsen

Summary:

This paper provides a very thorough statistical analysis of ice core records evidencing DO events in the last glacial interval (mostly the NGRIP d18O record, but some comparisons with the GRIP record are performed), as well potential external forcing factors such as among others global ice volume (as inferred from benthic d18O) and atmospheric CO2 from Antarctic ice cores. The focus is on first-order variability, and the proposed piecewise-linear fit is a suitable method to extract features such as stadial

and interstadial durations, as well as warming and cooling rates. The study provides a comprehensive summary of statistical features associated with the DO cycles, that should prove useful for benchmarking past future modelling studies attempting to explain the DO cycles in terms of physical mechanisms. In this sense, this manuscript provides a very valuable contribution to the literature of DO variability.

The paper is written very well, the authors' reasoning is clear and easy to follow, and the statistical analyses are performed very carefully, with appropriate account of uncertainties of different kinds. I thus suggest publication as soon as the following, minor comments are addressed:

Comments:

1. In the abstract and at several more occasions, it is stated that the goal of this paper is to obtain a mechanistic understanding of the DO cycles. This is not true in my opinion, since only statistical features are reported. These features can be used for benchmarking modelling studies testing different mechanisms, but this is not done in this paper.

2. Similarly, I would suggest to remove the word "causal" from the abstract and remainder of the manuscript: Only statistical similarities are tested, and no conditioning is performed to infer conditional dependencies. Also, no dynamical models are used, which could provide some hints at actual causality.

3. I don't think that previous work on the DO cycles is sufficiently recognized by the authors. For example, in the introduction it is stated that there are no established theories of the DO cycles. This is not true, there's a multitude of competing hypotheses, which can be broadly divided into two classes, namely those focussing on AMOC changes induced by freshwater forcing (the works of Ganopolski and colleagues and Timmermann in particular) and those focussing on sea ice changes (Li et al., J. Clim. 2010, Dokken et al., Paleooc. 2013, Petersen et al., Paleooc. , 2013, Boers et al., PNAS 2018, Sadatzki et al., Sci. Adv. 2019) In the last few years, DO-like oscillations have

been reproduced even in comprehensive models (Peltier & Vettoretti GRL 2014, Vettoretti & Peltier GRL 2013, Zhang et al., Nature 2014, Klockmann et al., 2019). A brief paragraph giving credit to these models seems in order.

4. Moreover, it is one of the key results of the study that global ice volume (inferred from benthic d18O) or temperature have a strong influence on the interstadial durations; this observation has, however, been made previously: Mitsui and Crucifix (Clim Dyn 2017) show from a statistical point of view that including this forcing is supported by the data, and Boers et al. (PNAS 2018) use it explicitly to infer the interstadial cooling rate during interstadials.

5. The ultimate goal of this study is to provide the statistical basis for discriminating between different mechanisms to explain the DO events, but this comparison of different mechanisms is not performed. Do the statistical features you extract give some hints at which of the prominent hypotheses listed above (point 3) are more likely? It would be nice to include at least a discussion on this at the end, as it is somewhat promised in the beginning.

―――――――――――――――――――――――

---

## Referee Comment (RC2) · Anonymous Referee #2 · 18 Apr 2019

Review of Lohmann and Ditlevsen 2019

Lohmann and Ditlevsen (2019) evaluate abrupt DO variability in the high-resolution NGRIP d18O record using piece-wise linear fitting. They perform correlation studies of the various model parameters with other climate records.

I find it difficult to judge this paper. On the one hand, the work is carefully done, the analysis statistically and mathematically robust, and the paper is well written. One the other hand, after 31 pages of manuscript the reader has not really learned anything that was not already described in the literature. The authors confirm the 17-yr old result by Schulz (2002), briefly revisit the result by Buizert and Schmittner (2015), and

confirm some of their own conclusions from Lohmann and Ditlevsen (2018) using a new method. One new element (the idea that cooling rates control interstadial duration, a variation on the Schulz argument) is based on the fallacy that correlation proves causation. Moreover, I have some major conceptual problems with their approach, which I will outline below.

In balance, the paper in its current form contributes too little to warrant publication. After major revision (some suggestions below) it may be reconsidered.

(1) In deciding which events are stadials the authors rely completely on (Rasmussen et al., 2014) – R14 hereafter, while this is of course the best starting place, it does mean that the authors inherit all the assumptions and historical numbering conventions that are in that study.

DO cycles come along a wide spectrum of shapes and timings, and deciding what is a stadial (indicated with numbers in R14) vs. a cold sub-event (lettered with b, d, f) in R14) is a very arbitrary choice. This leads to some strange results from Lohmann and Ditlevsen, such as stadials that lasts only 20 years (P18 L24). From a climatic perspective, such short cold periods are more likely to be cold sub-events rather than true stadials.

The cause of the cold sub-events (lettered with b, d, f) is of course not well known, but most likely they represent outburst floods (e.g. from proglacial lakes) that temporarily increase N-Atlantic sea ice cover and cool Greenland, the 8.2 ka event is the posterchild for such events. They occur most frequently during periods of ice sheet decay (MIS4-MIS3 and MIS2-MIS1 transitions) supporting this interpretation. When is a cold period a sub-event and when a true stadial? R14 applies some criterion of baseline separation; this makes GI-24.1b a "cold event" and GS24.2 a "stadial", even though they appear nearly identical and GS24.2 may just be a larger cold event (bigger outburst flood?). In other "stadial" cases (GS 14, GS 23) baseline separation is not achieved and R14 calls them stadials only because they choose to adhere to historical

[Figure]

DO numbering (based on older low-res cores). Many DO events have a clear warming period at their end (7a, 8a, 12a) – could DO 13 be one of those? The difference is arbitrary.

DO 24, 17, 16, 15 each consist of 2 DO events in the Greenland-centric R14, but in for example the Hulu speleothem record and the Iberian margin SST these events are recorded as single (and not double) events. Could these be single events from the perspective of the overturning circulation, but separated events in Greenland due to regional effects (like freshwater)?

These sub-events are always the outliers in the scatter plots (5a, 6a, 6b), making this an interesting point to consider. The authors note that "Buizert and Schmittner (2015) lump each of the interstadials 24, 23, 21, 17, 16, 15 and 2 together into one event, even though they are comprised of two events." But are they two events? If we put 100 paleoclimate researchers in one room, my guess is that less than half would call GS 24.2 a true stadial (minimum requirement to support the claim that DO 24 is comprised of 2 events). The vote may be different for GS 15.2. I don't want to claim here that they are 1 or 2 events, just that this depends completely on arbitrary definitions, and that reasonable people can disagree on the number of "true" DO cycles give that the cold sub-events are ubiquitous and can have a wide range of sizes.

One could turn the question on its head, and argue that, based on the fact that the sub events of DO 24, 23, 21, 17, 16, 15 consistently show up as outliers in scatter plots, and fit the climatic trends when lumped together, they are actually single (and not double) events from the perspective of the global climate system and oceanic overturning circulation.

One way to make the present paper more interesting is to try out different definitions of a stadial (beyond adopting R14), and see what definition may minimize scatter in the plots. What do other climate archives like speleothems suggest these events looked like? Would a duration threshold (e.g. cold period longer than 300 years) be a better

way to define a stadial? There are probably good climatic reasons why DO timescales are linked to global climate markers (as argued by e.g. Schulz 2002 and Buizert and Schmittner 2015), so an approach of finding an objective (multi-proxy?) stadial definition that minimizes scatter is justified in my view.

I want to emphasize this is not a critique of R14 itself, which seeks to provide a consistent nomenclature for events and has succeeded in doing so. The problem arises when R14 is mistaken for an objective and meaningful decision on which events are "true" stadials – which was never the aim of that study. The author's algorithm has the liberty to change the timing of transitions, but not the number of events, and so does not challenge the R14 definitions.

To summarize an excessively long comment, the approach by the authors has a fundamental and tenuous assumptions that is neither acknowledged nor examined. They interpret the R14 beyond its intended use as a climatically meaningful distinction of which cold events are stadials and which are not. While the shape fitting is done with much mathematical rigor, these underlying assumptions will always limit the validity of their conclusions. Trying different definitions for stadials would be an interesting research direction, as well as different proxy archives.

(2) One of the main conclusions of the paper is that cooling rates "control" the interstadial durations (P14 L13, P16 L21 and elsewhere). They are only correlated, which does not prove any kind of causation or control; both could be controlled by a third parameter such as AMOC strength, $CO_2$ or SH temperature. This correlation was discovered by Schulz (2002), who actually argues for an ice volume control.

The authors further suggest that the "interstadial duration is determined as soon as the rate is established" and that the duration is "determined" a few hundred years after the onset. However, if both cooling rate and duration are controlled by a third parameter (like AMOC strength), this interpretation is strange. In my view, Fig. 5b simply reflects the amount of data needed to determine the slope in a noisy time series, and is not

some time scale on which the interstadial duration is somehow "determined" by the coupled climate system.

On a side note, I am uncomfortable with language of transitions being "determined" thousands of years in advance. Interstadial terminations occur because at that time interactions between components of the climate system are favorable for such a transition (including "noise" components). The climate system is not a decision-making entity that plans things centuries to millennia ahead. The word "predicted" seems more appropriate. So please revise.

If the authors want to argue for their mechanism (cooling rate controls duration) they will have to provide a meaningful climatic pathway for such control, which is currently lacking. At the very least they have to clarify all the language suggesting causation.

(3) The discussion section is basically a lengthy summary of the preceding chapters, which is not the function of a discussions section. This will need to be rewritten. There are many caveats and assumptions that need to be addressed, and the work can be placed in a broader paleoclimatic context.

(4) In earlier work, Ditlevsen suggested that the DO transitions are purely noise-driven – others have probably made that suggestion also (Ditlevsen et al., 2007; Ditlevsen et al., 2005). Given that event durations are clearly correlated to global climate parameters, is that still your view?

(5) The referencing of published material is very minimal for a paper of this length on a topic that is so extensively written on. Much is in fact known about the DO cycle (despite the author's claims to the contrary). Marine sediment data clearly show a link to the Atlantic ocean circulation (see e.g. review by Lynch-Stieglitz, 2017), and climate modeling studies clearly implicate sea ice cover in the North-Atlantic. Many remote teleconnections have been clearly described, and several drivers have been proposed. Also, many more papers have used similar statistical techniques on the DO cycle that should be referenced.

(6) The authors ignore Heinrich events, while it is commonly believed that H-events lengthen the stadials in which they occur by putting freshwater into the North Atlantic. Please discuss this in the stadial duration section.

(7) The paper is very long and could be shortened substantially.

(8) There is no data availability statement.

(9) Are the d18O data corrected for mean ocean d18O? This would of course influence the cooling rates of long interstadials.

(10) The work mostly just confirms earlier work. I would encourage the authors to think about ways to broaden or improve the scope, to reward the reader with something new.

(11) The uncertainties in the fitting parameters are carefully estimated. Could they be listed in table 2? I imagine that for the very short interstadials (<400 yrs) things like the gradual cooling slope are not well constrained.

(12) Why do you use a constant stadial level? Do the stadial levels resemble Antarctica, as suggested by (Barker and Knorr, 2007).

(13) Have you tried including other records? The Ca record has much better signal to noise, allowing for better timing determination. I think marine and speleothem records have much to add to the problem also.

Comments on the text:

P1L7: remove "mechanistic". This is not attempted, in my view. Climate dynamics are not discussed.

P1L14: "largely unknown": A lot is known, the gist of which could easily be summarized in a few sentences.

P2L16: "we do not have to rely on any subjective choice of stadial and interstadial onset or levels". This is a misrepresentation, in my view. The subjective choices were

all made by

R14. The algorithm does not have the ability to decide independently how many events there are, and whether individual cold periods are stadials or just cold sub-events.

P2L24: "In contrast...state" But the transitions are not purely noise-driven, since the period durations are linked to global climate, no?

P3L13: Do the short cold events influence the fitting?

P4L29: Instead, we use ... basin-hopping. This means very little to much of your readership. Please elaborate or leave out.

P5L8: "climate features" should be "d18O features"

P5L12: "mechanistic" I don't think this is the right word. Climate dynamics are barely discussed.

P7L1: but you use 90% confidence, correct? How many false positives for 0.9?

P7 section 3: why is DO 1 omitted?

Fig 2: can you comment on DO 23.1? You routine picks its termination 3 ka before R14 does.

Section 4.1.1: Is there any conceivable mechanism by which cooling rates determine interstadial duration? Correlation and causation are falsely equated here.

P13 L13: What is lambda? Wasn't the cooling rate called s_2? Also specify whether you're talking about the gradual or fast cooling rate

P14L3: fixed cooling rate... You mean all interstadials would have the same cooling rate? We know this to be untrue. I do not understand this hypothetical scenario.

P14L8: "strong control". Correlation is not causation

P14L22: log-normal: is this meaningful since you only fit the tail of the log-normal

distribution?

P15L10: "determined" change wording. The climate system does not plan ahead millennia. If you want to argue for such a mechanism you should at least provide a dynamical pathway, even if speculative.

P16 L19: Not necessarily. Both duration and cooling rate are controlled by heat transport and interactions within the climate system, and appear to correlate to a third parameter like $CO_2$, ice volume, or similar. Within a few hundred years you can detect the cooling trend within the noisy time series, and because of the correlation you can predict the interstadial duration with reasonable accuracy at that time. Nothing is "determined" a few hundred years into the event.

P16L21: rewrite.

P18L16: see comments above. Whether these are one or two events depends on ones definition of a stadial. There is no widely accepted definition, and R14 provides nomenclature only and is not the final authority on this matter. Other archives should also weigh in on this question, since Greenland may reflect regional effects.

P18L24: a 20yr stadial is probably not a stadial in most people's definition. A time threshold (250 yrs?) may be of use in defining stadials. 250 yrs provides a rough timescale of Atlantic overturning (volume divided by rate), and makes some intuitive sense for that reason.

P18L27: the data ARE consistent with... (data are plural).

Fig 7a: all outliers are in MIS5. Is that relevant?

P19L6: common forcing makes most sense, right? Weak AMOC means low temps and long stadials?

Discussion: This is just a lengthy summary. Please discuss the strengths and shortcomings of your method, and place it in a broader context.

[Figure]

P24L5: "cooling rates clearly determine" rewrite

References:

Barker, S., Knorr, G., 2007. Antarctic climate signature in the Greenland ice core record. Proc. Natl. Acad. Sci. U. S. A. 104, 17278-17282.

Buizert, C., Schmittner, A., 2015. Southern Ocean control of glacial AMOC stability and Dansgaard-Oeschger interstadial duration. Paleoceanography 30, 2015PA002795.

Ditlevsen, P.D., Andersen, K.K., Svensson, A., 2007. The DO-climate events are probably noise induced: statistical investigation of the claimed 1470 years cycle. Clim. Past. 3, 129-134.

Ditlevsen, P.D., Kristensen, M.S., Andersen, K.K., 2005. The recurrence time of Dansgaard-Oeschger events and limits on the possible periodic component. J. Clim. 18, 2594-2603. Lohmann, J., Ditlevsen, P.D., 2018. Random and externally controlled occurrences of Dansgaard–Oeschger events. Clim. Past 14, 609-617.

Lynch-Stieglitz, J., 2017. The Atlantic Meridional Overturning Circulation and Abrupt Climate Change. Annual Review of Marine Science 9, 83-104.

Rasmussen, S.O., Bigler, M., Blockley, S.P., Blunier, T., Buchardt, S.L., Clausen, H.B., Cvijanovic, I., Dahl-Jensen, D., Johnsen, S.J., Fischer, H., Gkinis, V., Guillevic, M., Hoek, W.Z., Lowe, J.J., Pedro, J.B., Popp, T., Seierstad, I.K., Steffensen, J.P., Svensson, A.M., Vallelonga, P., Vinther, B.M., Walker, M.J.C., Wheatley, J.J., Winstrup, M., 2014. A stratigraphic framework for abrupt climatic changes during the Last Glacial period based on three synchronized Greenland ice-core records: refining and extending the INTIMATE event stratigraphy. Quat. Sci. Rev. 106, 14-28.

Schulz, M., 2002. The tempo of climate change during Dansgaard-Oeschger interstadials and its potential to affect the manifestation of the 1470-year climate cycle. Geophys. Res. Lett. 29, 2–1-2–4.

---

## Author Comment (AC1) · 16 May 2019

We thank the reviewer for his/her evaluation of the manuscript and helpful comments. Below, we reply to all comments of the reviewer.

1." [. . .] aim is to obtain mechanistic understanding of the DO cycles. This is not true in my opinion, since only statistical features are reported. These features can be used for benchmarking modelling studies testing different mechanisms, but this is not done in this paper."

We agree that this is not the best wording to describe the goal of the paper. We did

not mean "mechanistic" in the sense of "physical mechanism", but rather "empirical", indicating how the properties of the cycles develop over time, as a function of previous cycles and forcings. We use the word "empricial" instead in the revised manuscript.

2. "[. . .] remove the word "causal" [...]. Only statistical similarities are tested, and no conditioning is performed to infer conditional dependencies. Also, no dynamical models are used, which could provide some hints at actual causality."

We agree that the word "causal" is problematic to use in our context and removed/replaced it throughout the manuscript.

3. "I don't think that previous work on the DO cycles is sufficiently recognized by the au- thors. [. . .] "

We agree that a paragraph summarizing previous hypotheses is helpful and included one at the beginning of the introduction.

4." [. . .] ice volume [. . .] strong influence on the interstadial durations; this observation has, however, been made previously: Mitsui and Crucifix (Clim Dyn 2017) show from a statistical point of view that including this forcing is supported by the data, and Boers et al. (PNAS 2018) use it explicitly to infer the interstadial cooling rate during interstadials."

We thank the reviewer for pointing this out. We are aware of these studies, but we don't discuss them in the manuscript, because we do not actually argue for a correlation of the interstadial durations and ice volume. Even though this is not discussed in the manuscript, we find that for the whole data set, the correlation is not significant at 95% (can be seen in Fig. 5), and the influence of ice volume on interstadial durations is mostly due to the long interstadials 23.1 and 21.1 occurring at low ice volume. This fact is then further obscured by the short 23.2 and 21.2 happening during the same period.

In the paper, we are presenting results on the cooling rates as a predictor of the interstadial durations, a connection of the cooling rates and forcings (such as ice volume), referring to earlier work (Schulz 2002), and argue that CO2 is actually a better predictor than ice volume.

5. "The ultimate goal of this study is to provide the statistical basis for discriminating between different mechanisms to explain the DO events, but comparison of different mechanisms is not performed. Do the statistical features you extract give some hints at which of the prominent hypotheses listed above (point 3) are more likely? It would be nice to include at least a discussion on this at the end, as it is somewhat promised in the beginning."

We agree that this is desirable and reworked large parts of the Discussions Section in order to establish a stronger connection of our work and leading hypotheses or model experiments concerning DO events. To test our results in more detail in future studies, more models, or at least more runs of existing models, under different forcing scenarios are needed.

---

## Author Comment (AC2) · 16 May 2019

We thank the reviewer for his/her careful evaluation of the manuscript, which helped to significantly improve the manuscript. Below, we reply to all comments of the reviewer, while splitting some longer comments into multiple paragraphs.

"[. . .] after 31 pages of manuscript the reader has not really learned anything that was not already described in the literature. The authors confirm the 17-yr old result by Schulz (2002), briefly revisit the result by Buizert and Schmittner (2015), and confirm some of their own conclusions from Lohmann and Ditlevsen (2018) using a new method. "

[Figure]

It is obviously our duty to convey the novelty more clearly to the reader. We hope that this comes through better in the revised manuscript. We clarify this by a more precise interpretation of our results, as outlined in the following comments.

We do not confirm all results by Schulz (2002), but on the contrary we critically examine Schulz' results by first doing an actual statistical analysis of the relationship of cooling rates and interstadial durations. Secondly, we show the limitations in time of the ice volume forcing hypothesis, as well as statistical limitations relating to simple common linear trends. Furthermore, we derive that atmospheric CO2 is a better predictor of the cooling rates. Similarly, we only recover the result by Buizert and Schmittner (2015) when discarding outliers. The outliers themselves require further interpretation, which relates to the comments further below.

"One new element (the idea that cooling rates control interstadial duration, a variation on the Schulz argument) is based on the fallacy that correlation proves causation."

We believe this is a misunderstanding of language, which should now be clear in the revised manuscript. We do, of course, not imply that correlation proves causation. We simply observe that the interstadial durations are predictable to a reasonable degree after several hundreds of years. The physical climate process causing this cannot be identified within our data analysis and the cooling rates are solely an indicator allowing us to measure this process. (See also comments to 2))

We will clarify this in all corresponding parts of the manuscript.

Here is a brief summary of what is in our opinion the novelty: a) We develop a model for objective characterization of the DO events observed in ice core records as noisy saw-tooth "shapes".

b) We classify the DO events based on their consistency with the saw-tooth shape.

c) We extract a variety of features based on the model parameters obtained from the record, and look for relationships in between features and forcings. The analysis is

more extensive than any previous efforts and may be used as a reference for future research. It is (partly) summarized in the heat map of figure S4. To highlight this better, we have moved it to the main text (Sec. 4).

d) The interstadial durations can be predicted reasonably well a few hundred years after GI onset. This is due to the gradual cooling being sufficiently linear and consistent over time. → Thus, a noise-induced scenario is unlikely for the interstadial-stadial transitions.

e) In the younger half of the glacial, atmospheric CO2 is a good predictor of the cooling rates, whereas we cannot establish a significant driver for the older half.

f) Even though there is correlation with the stadial levels and insolation, the stadial durations cannot be predicted accurately. When removing forcing influences, the variability remains exponential.

g) We extract warming durations from the record and analyze the distribution and temporal evolution (versus forcings). A comparison with the theory of reactive paths for stochastic multi-stable system shows that this corroborates results from the stadial durations: The stadial-interstadial transition is consistent with a noise-induced scenario.

h) The DO warming amplitudes are weakly predictable from the stadial levels. However, they are not significantly correlated with external forcings, such as ice volume as has been suggested previously.

1) "In deciding which events are stadials the authors rely completely on R14. [. . . .], the authors inherit all the assumptions and historical numbering conventions that are in that study. DO cycles come along a wide spectrum of shapes and timings, and deciding what is a stadial [. . .] vs. a cold sub-event [. . .] is a very arbitrary choice."

We fully agree that the careful analysis and the classification made in R14 that could be debated. It is not the focus of our paper to challenge this classification, which we assume to be a reasonable basis for our analysis.

More specifically, in R14 a threshold is needed to decide which events are sub-events and which not. However, in any conceivable scenario this would be necessary to pick events out of smaller-scale background variability. The R14 classification is based on three ice cores and two proxies and from the point of view of Greenland climate seems justified. As such, it is "Greenland-centric", which is discussed in the comments further below.

We mention these assumptions in the revised manuscript, and acknowledge that some of our results depend on them (Changes document p3 l28 – p4 l2 and p29 l2-14).

"... This leads to some strange results from Lohmann and Ditlevsen, such as stadials that lasts only 20 years (P18 L24). From a climatic perspective, such short cold periods are more likely to be cold sub-events rather than true stadials."

The 20 years of GS-24.2 only refer to the time where the proxy is stable. The overall break in between adjacent interstadials (including cooling and warming) is about 170 years, and certainly more significant in amplitude than any sub-event in the Greenland records.

If driven by the AMOC changes, which our analysis does not assume a priori, the cycle of shutdown and resurgence would then be 170 years, which is close to the 250 years suggested by the reviewer. The numerical value of 20 years is unlikely to be important since we mostly use rank statistics in our analysis.

"... The cause of the cold sub-events [...] is of course not well known, but most likely they represent outburst floods (e.g. from proglacial lakes) that temporarily increase N-Atlantic sea ice cover and cool Greenland, the 8.2 ka event is the posterchild for such events. They occur most frequently during periods of ice sheet decay (MIS4-MIS3 and MIS2-MIS1 transitions) supporting this interpretation. When is a cold period a sub-event and when a true stadial? R14 applies some criterion of baseline separation; this makes GI-24.1b a "cold event" and GS24.2 a "stadial", even though they appear nearly identical and GS24.2 may just be a larger cold event (bigger outburst flood?). In other

"stadial" cases (GS 14, GS 23) baseline separation is not achieved and R14 calls them stadials only because they choose to adhere to historical DO numbering (based on older low-res cores). Many DO events have a clear warming period at their end (7a, 8a, 12a) – could DO 13 be one of those? The difference is arbitrary."

As above, this is more a discussion of the quality of the classification in R14, which we do not discuss, however, we have the following remarks:

In terms of Greenland ice cores, GI-24.1b and GS-24.2 are not nearly identical. Estimated form the high-resolution NGRIP d18O record, GS-24.2 lasts 170 years (when including warming and cooling periods) at an amplitude of ∼3.8 permil, whereas GI-24.1b is 130 years at ∼2.0 permil amplitude.

We agree that the case of GS-14 and GS-23 (labeled quasi-stadials by R14) is more debatable. First, if one compares GI-13 and GI-22 with the other "rebound" type events, they are simply more pronounced (longer, slightly larger amplitude). Second, we need to include them because only like this we can get a satisfactory saw-tooth fit throughout the record, and representative features for each event. This is very pronounced for the case of GS-23 (see Figure 1, where we merge GI-22 and GI-23.1 in the fit), and to a lesser extent for GS-14. We do not claim that our fit is a method to decide whether they should be considered individual events, but in order for our analysis to be meaningful, we require a reasonable fit.

While we admit that the choice of GS-14 and GS-23 might go one or the other way, since we do statistical robustness tests, our results do not rely on the presence or absence individual stadials. In fact, we performed a semi-complete analysis for a fit without using GS-23 (not discussed in the manuscript), yielding qualitatively unchanged results.

We appreciate the reviewers comment and discuss these issues in Section 2.1 of the revised manuscript (Changes document p3 l28 – p4 l2).

"... DO 24, 17, 16, 15 each consist of 2 DO events in the Greenland-centric R14, but in for example the Hulu speleothem record and the Iberian margin SST these events are recorded as single (and not double) events. Could these be single events from the perspective of the overturning circulation, but separated events in Greenland due to regional effects (like freshwater)?"

These sub-events are always the outliers in the scatter plots (5a, 6a, 6b), making this an interesting point to consider. The authors note that "Buizert and Schmittner (2015) lump each of the interstadials 24, 23, 21, 17, 16, 15 and 2 together into one event, even though they are comprised of two events." But are they two events? If we put 100 paleoclimate researchers in one room, my guess is that less than half would call GS 24.2 a true stadial (minimum requirement to support the claim that DO 24 is comprised of 2 events). The vote may be different for GS 15.2. I don't want to claim here that they are 1 or 2 events, just that this depends completely on arbitrary definitions, and that reasonable people can disagree on the number of "true" DO cycles give that the cold sub-events are ubiquitous and can have a wide range of sizes.

We agree that it is still unclear whether successions of shorter events found in the Greenland ice cores are complete reorganizations of the large-scale climate system (such as AMOC shutdown and resurgence), in the same way as longer events are.

While the Iberian margin SST record is not high-resolution enough to resolve the shortest events, Asian monsoon speleothem records, including Hulu cave, now exist in sufficient resolution, and indeed resolve events such as GI-21.2, and the rapid successions of GI-17.2 to GI-15.1 (Cheng et al. 2016). The same goes for Alpine speleothem records, which have a very detailed similarity to the Greenland ice cores. The event succession of GI-15-17 is seen (Moseley et al. 2014) as full-amplitude DO events, suggesting that the centennial variability is not regional to Greenland. Similarly, events 23.2 and 21.2 are identified (Boch et al. 2011).

Additionally, the short events are well recorded in methane records from Greenland and

Antarctica (WAIS Antarctic record: Rhodes et al. 2015, Rhodes et al. 2017; Greenland NEEM record: Chappellaz et al. 2013).

Of course, we cannot argue in this way that chosen events are indeed the most prominent globally. However, this problem is very hard to address, and beyond the scope of this work. A more 'global' assessment of which centennial- to millennial-scale abrupt climate changes should be considered most important is difficult since different regions and proxies are sensitive to a different degree to the various parts of the climate system undergoing change, and thus will most likely highlight different events. For example, Asian monsoon proxies are much more sensitive to Heinrich events, so it is difficult to use them to study the statistics of DO events. An assessment including different types of proxies in different regions will require a (subjective?) weighing of some kind to extract the type of abrupt climate change one wishes to study. Furthermore, dating issues will need to be overcome.

Since the Greenland ice cores are in a region that is very sensitive to the North Atlantic climate they are ideally suited to study climate changes related to the AMOC.

Nevertheless, we agree with the reviewer that shorter events might be more specific to Greenland and could have a different trigger than longer events that are more clearly recorded in other archives. Simply due to their shortness it is not unlikely that they indeed do not represent global reorganization of the oceanic circulation or climate system.

Including the short events in the analysis is a way of identifying how they are different, in terms of their detailed features as well as in the context of forcing. Indeed, the fact that certain events show up as outliers might be because they are caused by something different than the majority of events.

We acknowledge the open problem of the cause and significance of the shorter DO events in the Discussion Section of the revised manuscript (Changes document p29 l2-14).

"... One could turn the question on its head, and argue that, based on the fact that the sub events of DO 24, 23, 21, 17, 16, 15 consistently show up as outliers in scatter plots, and fit the climatic trends when lumped together, they are actually single (and not double) events from the perspective of the global climate system and oceanic overturning circulation."

We agree that the evolution of the properties of DO cycles over time and their relationship to external influences becomes clearer when certain shorter Greenland events are discarded. Still, we do not believe that if discarding outliers (or combining short events) the correlation with external forcings or climatic trends is strong enough to warrant such a reasoning.

"... One way to make the present paper more interesting is to try out different definitions of a stadial (beyond adopting R14), and see what definition may minimize scatter in the plots. What do other climate archives like speleothems suggest these events looked like? Would a duration threshold (e.g. cold period longer than 300 years) be a better way to define a stadial? There are probably good climatic reasons why DO timescales are linked to global climate markers (as argued by e.g. Schulz 2002 and Buizert and Schmittner 2015), so an approach of finding an objective (multi-proxy?) stadial definition that minimizes scatter is justified in my view."

We do not believe that it is good to put a constraint of stadial durations, such as 300 years, within our framework. We treat the record without preconceived notions of the underlying driver, i.e., AMOC changes or hypothesized outburst floods for short events, which has both merits and shortcomings.

In terms of multiple hypothesis testing it is risky to try out many different definitions of stadials and correlate them with many different global climate markers, in order to find a match. We would furthermore need to decide which features would we like to be explained by the global climate markers, which is unclear a priori.

Finally, doing a full analysis as presented in the paper with many different stadial definitions is unfortunately beyond our present capacity and is left for future work.

" ...  I want to emphasize this is not a critique of R14 itself, which seeks to provide a consistent nomenclature for events and has succeeded in doing so.  The problem arises when R14 is mistaken for an objective and meaningful decision on which events are "true" stadials – which was never the aim of that study. The author's algorithm has the liberty to change the timing of transitions, but not the number of events, and so does not challenge the R14 definitions.

To summarize an excessively long comment, the approach by the authors has a fundamental and tenuous assumptions that is neither acknowledged nor examined. They interpret the R14 beyond its intended use as a climatically meaningful distinction of which cold events are stadials and which are not. While the shape fitting is done with much mathematical rigor, these underlying assumptions will always limit the validity of their conclusions.  Trying different definitions for stadials would be an interesting research direction, as well as different proxy archives."

In our judgment, solving the problem of unique classification of DO events will require a much better globally extended set of high resolution, well dated climate records, which we do not have today and probably also a realistic theory of their cause. We consider this as a fundamental problem, but somehow disconnected from our results, which will hopefully be a steppingstone towards solving the puzzle of the DO events.

In the revised manuscripts, the assumption of a predefined set of DO events is acknowledged and discussed (Changes document p3 l28 – p4 l2 and p29 l2-14). An extension of our algorithm to also objectively detect events would be less powerful than the assessment of R14, because it is based on 3 ice cores and 2 proxies, which is why we did not design the algorithm in such a way.

We agree that the use of other proxy archives is very promising. However, within the framework of our method these are difficult to incorporate, since they do not show a DO signal that is consistent enough, and has comparable resolution and age control.

[Figure]

2) "One of the main conclusions of the paper is that cooling rates "control" the intersta-
dial durations (P14 L13, P16 L21 and elsewhere). They are only correlated, which does
not prove any kind of causation or control; both could be controlled by a third parameter
such as AMOC strength, $CO_2$ or SH temperature. This correlation was discovered by
Schulz (2002), who actually argues for an ice volume control.

The authors further suggest that the "interstadial duration is determined as soon as the
rate is established" and that the duration is "determined" a few hundred years after the
onset. However, if both cooling rate and duration are controlled by a third parameter
(like AMOC strength), this interpretation is strange. In my view, Fig. 5b simply reflects
the amount of data needed to determine the slope in a noisy time series, and is not
some time scale on which the interstadial duration is somehow "determined" by the
coupled climate system. "

We have tried to clarify this misunderstanding in the manuscript. If we may be a little
impertinent: If you are walking to a town 10 km away at a pace of five km/h, you'll be
there in two hours. I can predict that as soon as I know your pace. That may take me
five minutes to do, but that implies neither that five minutes has any significance for the
prediction (two hours), nor whether your speed and the distance to town are controlled
by any third parameter. The only "control" is that the speed -kept constant- controls the
arrival time.

Thus, the word "control" was meant in the context of our statistical analysis. As elabo-
rated in the manuscript, each interstadial duration is defined by the cooling amplitude
divided by the rate. If either the amplitudes or the rates have a clearly dominant vari-
ability they also effectively "control" the durations of the interstadials. We show that this
is the case for the rates. We rewrote parts of the manuscript to clarify this (Changes
document p17 l24-29, p18 l11-12 and p27 l1-8).

We agree with the reviewer that a mutual control by a third parameter is likely, which
is also what we are investigating extensively in the paper. In fact, we think it is not

only likely but necessarily the case. The gradual cooling rate is merely an indicator of the global climate reorganizing on a specific timescale. The termination of the interstadial, and thus its observed duration, is the final consequence of it. Both durations and cooling rate are driven by the same process, even though the actual interstadial termination might be governed by additional processes that influence the climatic threshold. Our empirical finding is that this timescale seems to be consistently expressed via the linear cooling already early on in the glacial. As a result, the end of the interstadial is already anticipated and can be predicted. We thus obtain the novel result that the interstadial-stadial transition is not purely noise-induced.

We agree with the reviewer that our result of the time elapsed until the durations can be predicted (150y-350y) is influenced by the amount of data needed to determine the slope. However, this time elapsed is only relevant because the cooling trend is actually "linear enough", which is not obvious a priori and which is what we implicitly test for.

" ... On a side note, I am uncomfortable with language of transitions being "determined" thousands of years in advance. Interstadial terminations occur because at that time interactions between components of the climate system are favorable for such a transition (including "noise" components). The climate system is not a decision-making entity that plans things centuries to millennia ahead. The word "predicted" seems more appropriate. So please revise."

We are unsure what is the basis for this argumentation. Concerning the climate system planning ahead: Some models would suggest that the DO cycles are self-sustained oscillations, with a period potentially slowly modulated by external forcing. In this case, interstadial terminations would indeed be determined centuries/millennia in advance due to the periodicity. We do not argue that this scenario is the case, but only want to point out that it is conceivable that climate transitions are determined ahead due to slow, deterministic processes that may be measured a long time before the actual transition.

Our use of the word "determined" is simply based on the observation that 1. the gradual cooling is sufficiently linear and 2. the variability of the rates is larger than of the amplitudes. As a result, the interstadials are reasonably well determined already a few hundred years after interstadial onset.

" ... If the authors want to argue for their mechanism (cooling rate controls duration) they will have to provide a meaningful climatic pathway for such control, which is currently lacking. At the very least they have to clarify all the language suggesting causation."

As above, we hope we can clarify this with better terminology and language in the revised manuscript. We do not aim to find a mechanism by which the cooling rates control the durations. It simply follows from our analysis that there is a time scale in the climate system, which manifests itself in the rate of the roughly linear interstadial cooling, which is established soon after interstadial onset, and which approximately predicts the duration of the interstadial.

We do look for an actual control among the external forcings and find that CO2 is the best predictor for the younger half of the glacial. However, whether CO2 really acts as a forcing remains to be seen as it also shows millennial-scale variability in line with the (Antarctic) DO cycle, and is likely a response to it. The mechanism by which CO2 influences interstadials depends on what drives the DO cycles in the first place, which is not known a priori and is hard to establish from our analysis.

3) "The discussion section is basically a lengthy summary of the preceding chapters, which is not the function of a discussions section. This will need to be rewritten. There are many caveats and assumptions that need to be addressed, and the work can be placed in a broader paleoclimatic context."

We agree and rewrote large parts of this section.

4) "In earlier work, Ditlevsen suggested that the DO transitions are purely noise-driven

– others have probably made that suggestion also (Ditlevsen et al., 2007; Ditlevsen et al., 2005). Given that event durations are clearly correlated to global climate parameters, is that still your view?"

In this study, we regard the warming and cooling transitions separately. Our analysis shows that neither the stadial nor the interstadial durations can be clearly matched unambiguously with a climate forcing over the entire glacial. Only the general trends may be explained by forcing, an exception being evidence for good correlation of $CO_2$ and the interstadial cooling rates in the younger half of the glacial.

Conceptually, evidence for a modulation of the occurrence frequency of events over time does not exclude that the individual transitions are purely noise-driven. The modulation only affects the expected value of the occurrence frequency. Nevertheless, this study shows that the interstadial-stadial transition appears to be less consistent with a noise-induced process, due to their predictability. On the other hand, we provide new independent evidence for a noise-induced scenario for the stadial-interstadial transition by comparing the DO warming durations to reactive trajectories. Of course, this is not the end of the story since new data/analysis could find evidence to the contrary.

5) "The referencing of published material is very minimal for a paper of this length on a topic that is so extensively written on. Much is in fact known about the DO cycle (despite the author's claims to the contrary). Marine sediment data clearly show a link to the Atlantic ocean circulation (see e.g. review by Lynch-Stieglitz, 2017), and climate modeling studies clearly implicate sea ice cover in the North-Atlantic. Many remote teleconnections have been clearly described, and several drivers have been proposed. Also, many more papers have used similar statistical techniques on the DO cycle that should be referenced."

We agree that a review of published hypotheses and model experiments is useful and added a paragraph at the beginning of the introduction. If the reviewer is aware of more papers with similar techniques that are relevant to our results, we are happy to include

them.

6) "The authors ignore Heinrich events, while it is commonly believed that H-events lengthen the stadials in which they occur by putting freshwater into the North Atlantic. Please discuss this in the stadial duration section."

We thank the reviewer for pointing this out. Indeed, we omitted a discussion of Heinrich events, because it is difficult to find manifestations of them in the Greenland records, and due to difficulties of timing them relative to Greenland ice cores. Nevertheless we did some statistical tests and included the following as a new sub-section (4.2.3) in the manuscript:

Besides DO events, Heinrich events are the other major mode of millennial-scale climate variability during the last glacial period. These events correspond to massive discharges of ice rafted debris found in ocean sediment cores (Heinrich, 1988), with large climatic impacts that are well-documented in numerous proxy records at various locations. While their duration and timing needs to be better constrained, we follow Seierstad et al. (2014) for the temporal link of Heinrich events and the GICC05 chronology. This yields the set of Heinrich events H2, H3, H4, H5a, H5, H6, H7a, H7b and H8, which overlap with stadials 3, 5.2, 9, 13, 15.1, 18, 20, 21.1 and 22, respectively. Since some of these Heinrich events might be less established in the community, we also look at the reduced set of H2, H3, H4, H5 and H6. We test whether these 'Heinrich stadials' have significantly different properties than the remaining stadials, such as longer durations, by randomly sampling 9 stadials (5 for the reduced set) from the entire set without replacement and calculating the mean duration of this subset. We repeat many times until we can estimate the probability of trials yielding a higher mean duration than the actual set of 'Heinrich stadials'. If this is less than 5% (corresponding to p=0.05) we reject the hypothesis that 'Heinrich stadials' have the same mean duration as the remaining stadials at 95% confidence. This test gives essentially the same results as a one-sided t-test.

[Figure]

For the full (reduced) set of Heinrich events we find p = 0.028 (p = 0.022). It is not obvious whether this should be considered significant in the sense of a hypothesis that Heinrich events prolong stadials. A better statistical test is needed, since if the events were to occurr randomly during the course of stadials they would naturally be found preferentially in longer stadials. We leave a resolution of this for upcoming work. Based on the idea that 'Heinrich stadials' are colder than normal, we perform a test on the stadial levels, yielding p = 0.052 (p = 0.047). Again, this is probably not significant since Heinrich events mostly occur in the younger glacial half with generally lower levels. We can reject the notion that DO events following Heinrich events are 'stronger'. A statistical test on the DO warming amplitudes yields p = 0.102 (p = 0.472), whereas a test on the interstadial durations yields p = 0.403 (p = 0.583). This might depend on the precise timing of H3, which in our analysis precedes the especially weak GI-5.1.

7) "The paper is very long and could be shortened substantially."

We tried our best, but believe that most of the content needs to remain in order to support our results and demonstrate statistical significance.

8) "There is no data availability statement."

Will do in the final version.

9) "Are the d18O data corrected for mean ocean d18O? This would of course influence the cooling rates of long interstadials."

The data are not corrected for mean ocean d18O. We agree that doing so will in principle influence the cooling rates. However, the changes in mean ocean d18O during the longest interstadial (GI-23.1) are only about 0.2 permil (Shackleton 2000; Waelbroeck et al. 2002), which is an order of magnitude smaller than the Greenland isotopic changes, and thus negligible for our analysis.

10) "The work mostly just confirms earlier work. I would encourage the authors to think about ways to broaden or improve the scope, to reward the reader with something

new."

With our response comment (specifically the beginning and 1) and 2)) and corresponding revisions to the manuscript, we hope to convince the reviewer that our paper does not merely confirm earlier work, but instead critically assesses it, and furthermore provides a number of new results.

11) "The uncertainties in the fitting parameters are carefully estimated. Could they be listed in table 2? I imagine that for the very short interstadials (<400 yrs) things like the gradual cooling slope are not well constrained."

It is hard to list all uncertainties in this table, since we preferentially need two numbers to specify the uncertainties in terms of 1-sigma ranges. This would result in a very squeezed table.

Regarding the cooling slopes of shorter interstadials: Uncertainty in the rates needs to be assesses with care. We propose to look at the ratio of the 1-sigma range to the best fit value. We find that indeed the cooling rates of the shorter interstadials are more uncertain than those of the longer interstadials. There is a significant negative correlation (Spearman r = -0.74) of the relative uncertainty in the rates and the duration, as expected due to the smaller amount of data available to estimate the slope for shorter events. See Figure 2 showing the relative uncertainty in the slope vs. duration.

There are two events that we indeed could say are not well constrained: GI-17.2 and 15.2. We also exclude these from our analysis investigating the predictability of interstadials. (Sec 4.1.3)

In the manuscript, error bars for the logarithmic cooling rates can be seen in Fig. 6 c/d. Taking into account the many orders of magnitude of the cooling rates, we conclude that apart from the two outliers (can be seen as largest error bars in the plot), the other cooling rates are well constrained, as can be seen by the error bars on the lgarithmic scale. Simply put, fast rates are fast even when taking into account the error, which in

some cases can be more than 100% relative.

12) "Why do you use a constant stadial level? Do the stadial levels resemble Antarctica, as suggested by (Barker and Knorr, 2007)."

With few exceptions, the stadials do not show a clear trend, but rather fluctuate around a mean. For the purpose of estimating the stadial or interstadial onset times, using a constant stadial part works sufficiently good. Looking into the finer details of trends within the stadials is certainly of interest, but was beyond the scope of our paper.

We have no reason to believe that the Greenland d18O stadials resemble the Antarctic record, as observed by Barker and Knorr for a detrended/dejumped Greenland dust record. The constant stadial levels do correlate with the Antarctic EDML record at stadial onset, however this is mostly due to common linear trend, as mentioned in Section 4.2.2. A more convincing correlation is found with insolation.

13) "Have you tried including other records? The Ca record has much better signal to noise, allowing for better timing determination. I think marine and speleothem records have much to add to the problem also."

We have not tried to include records other than Greenland d18O records. The Greenland Ca record has a more complicated shape that is not as well captured by a piecewise-linear fit throughout the whole glacial. We agree that marine and speleothem records are important to answer further questions relating to DO events. However, as mentioned in our comments to 2), they do not have such a consistent DO signal (which there are probably good reasons for). Furthermore, they are less ideal in terms of resolution and timescale. The layer-counted Greenland ice core records provide excellent relative time constraints allowing for the determination of stadial, interstadial, and abrupt warming durations.

P1L7: remove "mechanistic". This is not attempted, in my view. Climate dynamics are not discussed.

Ok, we use the word "empricial" instead.

P1L14: "largely unknown": A lot is known, the gist of which could easily be summarized in a few sentences.

We agree and added a new opening paragraph with some references to published hypotheses and modeling work.

P2L16: "we do not have to rely on any subjective choice of stadial and interstadial onset or levels". This is a misrepresentation, in my view. The subjective choices were all made by R14. The algorithm does not have the ability to decide independently how many events there are, and whether individual cold periods are stadials or just cold sub-events.

We agree that the sentence as it is can be misleading. It was meant to refer to choices of what constitutes an event, but to choices of a segmentation of the record if we would ramp-fit each event individually. We reformulated and hope it is clearer now: "Thus, our results are not sensitive to subjective choices of cutting the record at predefined times before and after a transition."

P2L24: "In contrast. . .state" But the transitions are not purely noise-driven, since the period durations are linked to global climate, no?

We indeed find some correlation of the stadial durations with insolation. However, since the correlation is not very strong, we interpret this as a modulation in time of the expected residence time in the stadial state, which does not mean that the transitions themselves are not noise driven. In the paper we present that even if one removes the trend due to forcing on the stadial durations, we still find an exponential distribution, as expected for noise-induced phenomena.

If one considers a slow modulation of the average residence time as part of the driver, then the term 'purely' noise-driven might be problematic.

P3L13: Do the short cold events influence the fitting?

Although they might influence the fitting in some minor way, you can visually verify that they are mostly 'ignored' by the linear fit, and the fit just follows the linear slope of the remaining parts of the interstadial. (see e.g. GI-24.1, GI-17.1, GI-16.1, GI-14, GI-13)

P4L29: Instead, we use . . . basin-hopping. This means very little to much of your readership. Please elaborate or leave out.

We agree and omit the term here. Basin-hopping is explained in Appendix A, and a citation is given.

P5L8: "climate features" should be "d18O features"

Ok, we leave out 'climate'.

P5L12: "mechanistic" I don't think this is the right word. Climate dynamics are barely discussed.

We agree and use the word 'empirical' instead.

P7L1: but you use 90% confidence, correct? How many false positives for 0.9?

Indeed we mostly consider correlations etc. at 90% for further testing, and at 90% there are 45 false positives, which we added in the revised manuscript. The p<0.1 mentioned in the manuscript is just a guideline of how we proceed with this large number of statistical tests. Note that we often also investigate correlations that are not significant 90% for the entire data set, as mentioned earlier in the manuscript.

P7 section 3: why is DO 1 omitted?

As mentioned in the manuscript, GS-2.1 is non-stationary and would influence both GI-2.1 and GI-1 in a way that many features, such as warming amplitude and durations, would need to be discarded as outliers for these events. The d18O values during this stadial rise higher for a long period of time than the maximum of the preceding GI-2.1.

Fig 2: can you comment on DO 23.1? You routine picks its termination 3 ka before R14

does.

From the NGRIP d18O and the notion of stadials being defined as roughly constant low values after a gradual cooling this seems justified. In NGRIP, GI-23.1 consists of roughly 10kyr of gradual cooling, followed by 3.5kyr of approximately constant values leading up to the warming of GI-22. Since our analysis is only based on one ice core and proxy we however do not suggest that this is sufficient to argue for a definition of GI/GS-23.1, which is why we do not discuss this in the manuscript. Still, our analysis is unlikely to be sensitive to the precise parameters of a single event.

Section 4.1.1: Is there any conceivable mechanism by which cooling rates determine interstadial duration? Correlation and causation are falsely equated here.

We refer the reviewer to our previous comments on 2). Our analysis suggests that the cooling rates and their associated timescale are a manifestation of a deterministic climate process that pre-determines the interstadial durations. From our analysis we cannot establish the nature of this climate process. We adjusted the wording in this paragraph to not give the impression we equate correlation and causation.

P13 L13: What is lambda? Wasn't the cooling rate called s_2? Also specify whether you're talking about the gradual or fast cooling rate

We agree that this might cause confusion. It is a new definition that we added to make the notation more intuitive, along with D_I for the interstadial durations and A for the warming amplitudes. For consistency, we added these definitions to Table 1 in the revised manuscript.

P14L3: fixed cooling rate. . . You mean all interstadials would have the same cooling rate? We know this to be untrue. I do not understand this hypothetical scenario.

We are explaining the two limiting scenarios of either the amplitudes of the rates to be perfectly constant. Of course neither is true, but we test in the following which scenario is closer to the truth.

P14L8: "strong control". Correlation is not causation

We rephrased to the following: "Thus, from the perspective of linear interstadial cooling, the interstadial durations over the entire glacial are indeed largely governed by the cooling rates, in agreement with..."

P14L22: log-normal: is this meaningful since you only fit the tail of the log-normal distribution?

We are not only fitting the tail of the log-normal distribution. Maybe the reviewer refers to the AD test being tail-sensitive? In this case, we also use the Cramer-von Mises and Kolmogorov-Smirnov tests, which yield qualitatively unchanged results. We mentioned this too far below in Section 4.3.1, but in the revised manuscript moved this to Section 2.4, where it is more approriate.

P15L10: "determined" change wording. The climate system does not plan ahead millennia. If you want to argue for such a mechanism you should at least provide a dynamical pathway, even if speculative.

Changed sentence to: "If the rates govern the durations much more so than the cooling amplitudes, then the durations can already be approximately predicted as soon as the rate is established, which might happen early in the interstadial."

P16 L19: Not necessarily. Both duration and cooling rate are controlled by heat transport and interactions within the climate system, and appear to correlate to a third parameter like $CO_2$, ice volume, or similar. Within a few hundred years you can detect the cooling trend within the noisy time series, and because of the correlation you can predict the interstadial duration with reasonable accuracy at that time. Nothing is "determined" a few hundred years into the event.

We refer the reviewer to our response to Comment 2) above.

P16L21: rewrite.

"Given the previous result, we investigate..."

P18L16: see comments above. Whether these are one or two events depends on ones definition of a stadial. There is no widely accepted definition, and R14 provides nomenclature only and is not the final authority on this matter. Other archives should also weigh in on this question, since Greenland may reflect regional effects.

We refer the reviewer to our previous comments above. We now worded differently: "This disagreement comes from the fact that Buizert and Schmittner (2015) view each of the interstadials 24, 23, 21, 17, 16, 15 and 2 as one event, whereas we consider these as two events each, as suggested by the analysis of Rasmussen et al. (2014)."

P18L24: a 20yr stadial is probably not a stadial in most people's definition. A time threshold (250 yrs?) may be of use in defining stadials. 250 yrs provides a rough timescale of Atlantic overturning (volume divided by rate), and makes some intuitive sense for that reason.

See our response to Comment 1)

P18L27: the data ARE consistent with. . . (data are plural).

Ok.

Fig 7a: all outliers are in MIS5. Is that relevant?

We agree that this might be relevant and it holds not only for Fig 7a, but for all models we use for the stadial durations, as mentioned in the manuscript.

Since the correlation of the remaining events is not very strong but "the models fit only the overall trend and leave unexplained variability on top of it", we prefer not to interpret the fact that the outliers are in MIS5.

P19L6: common forcing makes most sense, right? Weak AMOC means low temps and long stadials?

We agree that it makes intuitive sense for long stadials to be also the coldest. However, we don't know whether this would be due to background forcing or as a result of the mechanism driving the climate transitions, such as AMOC changes, or both. From the perspective of our study the drivers and (most relevant) forcings are not known a priori, which is why we are testing evidence for it.

Discussion: This is just a lengthy summary. Please discuss the strengths and short-comings of your method, and place it in a broader context.

We agree and rewrote the Section.

P24L5: "cooling rates clearly determine" rewrite

We rewrote: "Because these durations are much more strongly governed by the cooling rates than by the cooling amplitudes they can be predicted to a good approximation as soon as the cooling rates have stabilized."

Literature

Boch, R. et al.: Clim. Past, 7, 1247–1259 (2011)

Buizert, C. and Schmittner, A.: Paleoceanography, 30, 1595–1612 (2015)

Chappellaz, J. et al.:Clim. Past, 9, 2579–2593 (2013)

Cheng, H. et al.: Nature, 534, 640-646 (2016)

Moseley, G. E. et al.: Geology, 42 (12), 1043–1046 (2014)

Rashid, H. et al. Paleoceanography, 18 (4), 1077 (2003)

Rasmussen, S. O. et al.: Quat. Sc. Rev., 106, 14–28 (2014)

Rasmussen, T. L. et al.: Paleoceanography, 18 (1), 1018 (2002)

Rhodes, R. H. et al.: Science, 348, 1016-1019 (2015)

Rhodes, R. H. et al.: Global Biogeochem. Cycles, 31, 575–590 (2017)

[Figure]

Sanchez Goni, M. F. and Harrison, S. P.: Quat. Sc. Rev., 29, 2823–2827 (2010)

Schulz, M.: Geophys. Res. Lett., 29, 1002 (2002)

Seierstad, I. K. et al.: Quat. Sc. Rev., 106, 29-46 (2014)

Shackleton, N. J.: Science, 289 (5486), 1897-1902 (2000)

Waelbroeck, C. et al.: Quat. Sc. Rev., 21, 295-305 (2002)

[Figure]

**Fig. 1.** Fit of GI-23.1 and GI-22 as one event

[Figure]

**Fig. 2.** Relative error of GI cooling rates vs their duration

[Figure]

---

## Author Response (AR2)

Dear Dr. Stenni,

thank you very much for reconsidering our manuscript. We revised the manuscript taking into account all of the referee's suggested modifications. We also shortened it by about 1.5 pages, but were unfortunately not able to shorten it any further without impacting the quality and completeness. The abstract is still up-to-date. Below you can find our answer to the referee comments.

Concerning the data availability, the principal data is not yet fully publicly available for the whole glacial period, but will be published soon by colleagues at Physics of Ice, Climate and Earth of the University of Copenhagen. Until then, the data can be requested from the authors. We updated the data availability statement accordingly. All relevant data generated by our study is published as part as the manuscipt, as well as the relevant algorithms that were written by the authors (in the appendix).

Best regards,
Johannes Lohmann

**Response to the referee:**

*"The authors take a different view on what the words "control" and "determine" mean than I do. The majority of readers will be interested in DO climate dynamics and physics rather than statistics, and so these words need to be changed to bring them into alignment with how the majority of readers would understand them. "*

In the revised manuscript, we do not use the word control anymore when talking about specific results. The word determined was to be understood as in "coefficient of determination". Nevertheless we replaced it according to suggestions by the referee.
Our paper is not a mere exercise in statistics. However, any reader with a climate dynamics and physics background should be able to follow our reasoning based on statistics, which after all is the foundation of empirical science.

*"The authors still confuse causation with correlation. In the old manuscript they claim that cooling rates "control" the interstadial durations – in reality they are merely correlated. In the revised manuscript they have simply changed the word "control" into "govern". This is unacceptable. Simply changing a word does not solve the underlying scientific fallacy. There is no mechanistic link given by which cooling rates control/govern the interstadial duration. More likely, both are controlled by a third parameter. The authors concede this point. This must be corrected, because both words imply (mechanistic) causality, which is not proven or even made plausible. The authors suggest that their words must be interpreted in a statistical context. I am note entirely sure what that means, but to the majority of readers this will not be how they interpret the word "govern". "*

As detailed in our previous response we are well aware of this potential pitfall and do not imply mechanistic causality from the rates to the durations, or any other correlation presented in the paper. Instead, we use the statistical analysis as a tool to uncover or suggest such potential causal relationships, which need to be checked elsewhere.
For the case of the interstadial rates and durations, we clarified the following interpretation of our statistical results in the revised manuscript (P25L19ff): "*We interpret this such that after interstadial*

*onset a large-scale reorganization of the climate system takes place on a timescale, which, even though very different from event to event, can be inferred from the cooling rate and stays consistent throughout the interstadial. We suggest that this reorganization is a major driving force of the DO cycle because its timescale predicts with reasonable accuracy when the interstadial-stadial transition takes place, which as a result cannot be purely noise-induced."*

So in our interpretation we only imply that there must be an underlying process that determines the interstadial durations, and this process is diagnosed by the cooling rate. Thus, it is should be clear that no direct causal link from the cooling rates to the durations is implied.

Similarly, we write in P17L12ff: *"Our interpretation is that the cooling rate is an indicator of a timescale of large-scale climate reorganization, which can already be measured relatively early in the interstadial and which remains approximately constant. Although we can see that there are exceptions, we conclude that for most events the interstadial duration can be predicted a few hundred years after the rapid warming."*

We furthermore changed passages containing the word "govern".

P25L16ff: Because these durations  correlate better with… they can be predicted to a good approximation as soon as the cooling rates have stabilized.

P16L17ff: If the  durations are correlated much stronger with the cooling rates than with the amplitudes, they can already be approximately predicted as soon as the rate is established, which might happen early in the interstadial.

*"The authors maintain that it is conceivable that the interstadial duration is "determined" hundreds to thousands of years ahead of time. The correct word here is "can be predicted". They give an example of walking towards a town at a fixed rate. My arrival time can be predicted once my speed is known, but it is not yet "determined". I could decide to take a 1 hour coffee break. The outcome of a democratic election can be \*predicted\* by polling the voters, it is not yet \*determined\* by this process. Likewise, once several hundred years have elapsed within an interstadial, the cooling rate can be estimated reliably, which means the duration of that interstadial can be \*predicted\* because the cooling rate is \*correlated\* with the duration (it does \*not\* control/govern it). This does not mean that the duration is already determined – volcanic eruptions, ice shelf collapse, or any number of things could happen that influence the true duration. "*

We agree that there are of course other factors that influence the true duration. We do not claim that the duration is absolutely, precisely determined from the rates, but from our statistical analysis it follows that they are the dominant determining factor, since they explain a very large part of the variance of the durations. The part of the stadial duration variance that cannot be explained by the rates estimated from 150-350 years after GI onset indeed reflects other processes that influence the durations. However, from our results it follows that for the majority of DO events these factors are negligible compared to the basic timescale set by the cooling rates shortly after interstadial onset.

We added in Sec. 4.1.3: *"Some of the unexplained variance of this prediction is due to other factors influencing the interstadial duration that are not diagnosed by the linear cooling rate, but, e.g., by the cooling amplitude."*

We also changed the wording in the Discussion: "*We suggest that this reorganization is a major driving force of the DO cycle because its time scale predicts with reasonable accuracy when the interstadial-stadial transition takes place, which as a result cannot be purely noise-induced.*"

"*The discussion section is still just a summary of the work, now with a single paragraph of true discussion added (last one). This is not the function of a discussion section.*"

We agree that some elements of a summary are present, which have been shortened significantly in the revised manuscript. But we would like to point out that there is more "discussion" than just the last paragraph. Also some of the individual results of our study need to be synthesized to draw conclusions, which we believe is most appropriate in the Discussion section. We hope to have improved the manuscript by the following restructuring of the Discussion section:
The first two paragraphs hightlight some possible shortcomings with the method and analysis.
The remaining paragraphs synthesize the individual statistical results and compare them with previous work, and interpret our results in the context of more realistic modeling and other data studies.

"*The authors have not attempted to shorten the paper. I still think this is appropriate, but I'll leave this up to the editor.* "

We have now shortened the paper by roughly 1.5 pages, which is unfortunately the best we can do without compromising the completeness of our analysis.

"*Data availability. The authors are not compliant with the CP data policy, which states that "Copernicus Publications requests depositing data that correspond to journal articles in reliable (public) data repositories, assigning digital object identifiers, and properly citing data sets as individual contributions." The 5yr data series should be publicly archived in a long-term database. These data have been used in several papers now, and requiring readers to contact a colleague of the authors (who is close to retirement age and not known for his reliability in responding to email) is not an acceptable form of archiving.*"

We agree that the data has to be made public and colleagues in Copenhagen are in the process of making the complete data set available together with a data description paper. We hope this to be finished within a few months. Until then, we suggest that researchers interested in the data contact the corresponding author. We have changed the data statement to clarify why the data is not publicly available yet.

[revised manuscript text omitted]